# Extreme suction attachment performance from specialised insects living in mountain streams (Diptera: Blephariceridae)

**Victor Kang**[1]*[†], **Robin T White**[2], **Simon Chen**[1], **Walter Federle**[1]

[1]Department of Zoology, University of Cambridge, Cambridge, United Kingdom; [2]Carl Zeiss Research Microscopy Solutions, Pleasanton, United Kingdom

**Abstract** Suction is widely used by animals for strong controllable underwater adhesion but is less well understood than adhesion of terrestrial climbing animals. Here we investigate the attachment of aquatic insect larvae (Blephariceridae), which cling to rocks in torrential streams using the only known muscle-actuated suction organs in insects. We measured their attachment forces on well-defined rough substrates and found that their adhesion was less reduced by micro-roughness than that of terrestrial climbing insects. In vivo visualisation of the suction organs in contact with microstructured substrates revealed that they can mould around large asperities to form a seal. We have shown that the ventral surface of the suction disc is covered by dense arrays of microtrichia, which are stiff spine-like cuticular structures that only make tip contact. Our results demonstrate the impressive performance and versatility of blepharicerid suction organs and highlight their potential as a study system to explore biological suction mechanisms.

**\*For correspondence:**
k.kang@imperial.ac.uk

**Present address:** [†]Department of Bioengineering, Imperial College London, London, United Kingdom

**Competing interest:** The authors declare that no competing interests exist.

## Introduction

Of the approximately one million known species of insects, only 325 attach using muscle-controlled suction organs (*Stork, 2018*; *Roskov et al., 2020*). These species belong to a single dipteran family, the Blephariceridae, and their larvae and pupae develop on rocks in torrential alpine streams where flow rates can exceed 3 ms⁻¹ (*Frutiger and Buergisser, 2002*; *Zwick, 2004*; *Figure 1a & b*; *Video 1*). Each blepharicerid larva has six ventral suction organs to attach to biofilm-covered rock surfaces, where it feeds on diatoms. Using its suction organs, the larva can locomote relatively quickly and possibly over long distances: blepharicerid larvae migrate from one stone to another to find the swiftest regions of the stream (*Frutiger, 1998*; *Mannheims, 1935*). Once development is complete, the winged adult emerges from its pupa, floats to the water surface, and immediately flies away to mate and lay eggs to begin the cycle anew (*Oosterbroek and Courtney, 1995*; *Craig, 1966*).

The remarkable morphology of blepharicerid suction organs is well described (*Mannheims, 1935*; *Rietschel, 1961*; *Kang et al., 2019*; *Komárek and Wimmer, 1922*). The organ superficially resembles a synthetic piston pump, with a suction disc that interacts with the surface and creates a seal, a central piston, and powerful piston muscles to manipulate the pressure, and a suction chamber with a thick cuticular wall to withstand low pressures during attachments. There are spine-like microstructures called microtrichia on the suction disc that contact glass surfaces and may increase resistance to shear forces. In addition, we have shown that a dedicated active detachment system allows the larva to rapidly detach its suction organ during locomotion (*Kang et al., 2019*).

While much is known about their morphology, the mechanisms involved in blepharicerid suction attachment are less well understood. Two studies to date have measured the attachment performance of blepharicerid larvae (*Frutiger, 2002*; *Liu et al., 2020*), yet neither of them offers mechanistic

**eLife digest** Suction cups are widely used to attach objects to surfaces in bathrooms and kitchens. They work well on tiles and other smooth surfaces, but do not stick well to rougher materials like brick or wood because they are unable to form an air-tight seal.

Researchers have been searching for ways to improve these cups by studying how octopuses, remora fish and other sea animals use muscle-powered suction organs to stick to wet and rough surfaces. However, the experiments needed to understand the detailed mechanics of suction organs are difficult to perform on living specimens of these animals.

The aquatic larvae of a family of insects known as the net-winged midges also have suction organs that are powered by muscles. These insects survive in fast flowing mountain streams where they use their suction organs to stick to rocks underwater. However, it remained unclear how these suction organs work.

Here, Kang et al. found that net-winged midge larvae attach extremely well to a variety of surfaces. The larvae were able to withstand forces over one thousand times their body weight when attached to smooth surfaces. Even on rough materials, where human-made suction cups attach poorly, the larvae were able to withstand forces up to 240-times their body weight.

Further experiments using several microscopy approaches revealed that the suction organs of the larvae are covered in multiple spine-like structures called microtrichia that interlock with bumps and dips on a surface to help the organ remain in place. Similar structures have previously been found on the suction organs of remora fish, but are not as tightly packed together.

These findings demonstrate that net-winged midge larvae may be useful model systems to study how natural suction organs operate. Furthermore, they provide a new source of inspiration for scientists and engineers to design and manufacture suction cups capable of attaching to a wider variety of surfaces.

insights into how their suction organs cope with different surface conditions to generate strong underwater attachments.

Suction is one of the main strategies for strong and controllable underwater adhesion. Biological suction organs can adhere with high strength, rapid controllability, and reusability on smooth, rough, and biofilm-covered surfaces (*Ditsche and Summers, 2014b*). This is in remarkable contrast to artificial suction devices widely used in technical applications, which allow only slow control and are limited to clean and smooth surfaces. Despite their potential for bio-inspiration, there are only a few well-studied animals (namely, remora fish, clingfish, octopus, and leeches), for which the function of specific structures in biological suction attachments has been experimentally demonstrated (*Beckert et al., 2015*; *Fulcher and Motta, 2006*; *Arita, 1967*; *Wainwright et al., 2013*; *Kampowski et al., 2016*; *Kampowski et al., 2020*; *Ditsche et al., 2014c*; *Kier and Smith, 1990*; *Kier and Smith, 2002*; *Smith, 1996*). To date, mechanistic studies on biological adhesion have focused primarily on terrestrial climbing animals such as geckos, tree frogs, insects, and spiders (*Lengerer and Ladurner, 2018*; *Federle and Labonte, 2019*), and have greatly expanded our knowledge on how to achieve and control adhesion in air. Likewise, mechanistic studies on biological suction are needed to identify new strategies for generating and controlling underwater adhesion in different surface conditions.

Here we have investigated the mechanisms underlying suction attachments of blepharicerid larvae. We first conducted a detailed morphological study of *Hapalothrix lugubris* (Blephariceridae) to provide new insights into structures that are relevant for suction attachments. To understand how well blepharicerid larvae attach to different surfaces, we quantified their performance on smooth, micro-rough, and coarse-rough surfaces. We compared blepharicerid suction performance with that of a model terrestrial insect to investigate how two fundamentally distinct adhesive systems cope with surface roughness. Finally, we examined the function of spine-like microtrichia through in vivo visualisation of the contact zone during attachments on smooth and microstructured substrates.

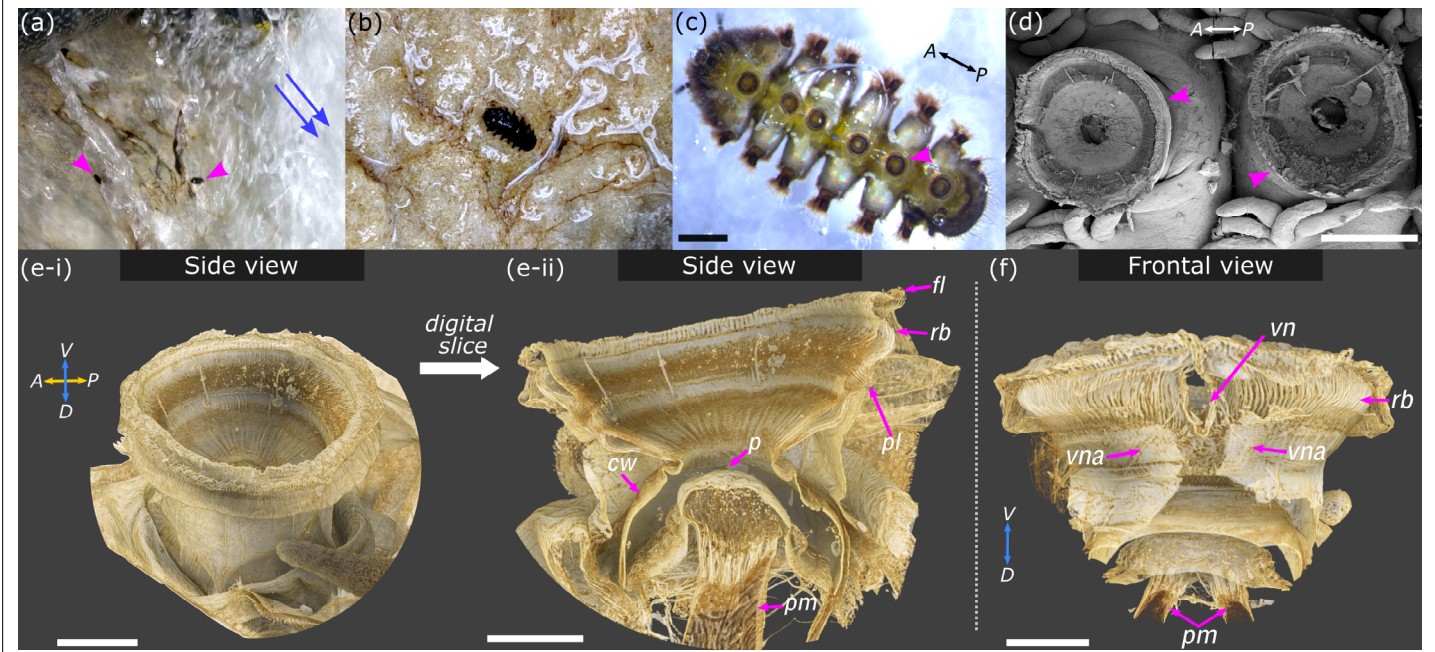

**Figure 1.** Overview of *Hapalothrix lugubris* and their suction attachment organs. (**a**) *Hapalothrix lugubris* larvae live attached to rocks in torrential alpine waterways. Blue arrows indicate stream flow direction. Arrowheads highlight two larvae revealed from a brief obstruction of the waterflow. (**b**) *H. lugubris* larva (dorsal view) on natural substrate. (**c**) Ventral view of a larva showing its six suction organs (one organ marked by arrowhead). (**d**) Scanning electron micrograph showing two suction organs (arrowheads). (**e-i**) Computed microtomography rendering of one whole organ. A: anterior, P: posterior, D: dorsal, V: ventral. (**e-ii**) Side view after digital dissection showing the following structures: outer radial beams (*rb*), palisade layer (*pl*), piston cone (*p*), and piston muscles (*pm*). The cuff wall (*cw*) encircles the suction cavity, and the outer fringe layer (*fl*) encircles the disc. (**f**) Frontal view showing the V-notch (*vn*) and its pair of apodemes (*vna*) extending dorsally into the body. Outer cuticle has been digitally dissected to reveal the radial beams. Note the pair of piston muscles extending dorsally. V: ventral, D: dorsal. Scale bars: (**c**) and (**d**) 500 µm; (**e-i**), (**e-ii**), and (**f**) 100 µm.

## Results

### Morphology of the suction attachment organ of *Hapalothrix lugubris*

*H. lugubris* larvae have six ventromedian suction organs, with each organ comprising a suction disc, a central opening and a piston, a suction chamber surrounded by a thick-walled cuticular cuff, and a V-notch (*Figure 1c-f* and *Video 2*). The suction disc contacts the surface for attachment, and the piston and underlying piston muscles (*Figure 1d & e*) actively lower the pressure inside the suction chamber. Two apodemes attaching to the V-notch in *H. lugubris* mediate its muscle-controlled opening for rapid detachment of the suction organ (*Figure 1f*; see also *Kang et al., 2019*).

The ventral disc surface of *H. lugubris* is covered in a dense array of microtrichia (*Figure 2a-e*). The suction disc-sealing rim, which seals the disc for suction attachment, closely resembles that of *Liponeura cinerascens* (*Kang et al., 2019*) and comprises a dense array of upright rim microtrichia (*Figure 2b*). This is different from *Liponeura cordata*, which has a distinct rim made up of a single row of horizontally flat rim microtrichia (*Kang et al., 2019*). Going from the rim to the centre of the disc, the short rim microtrichia transition into longer spine-like microtrichia (6.7 ± 0.5 µm in length and 0.56 ± 0.01 µm in mid-length diameter; mean of means ± standard error of the mean; measured from scanning electron microscopy (SEM) images of n = 2 individuals), and then again to shorter microtrichia in the centre.

The following imaging techniques were used to gain insights into the ultrastructure and internal organisation of the suction disc: freeze-fracture SEM, 3D models using computed microtomography (micro-CT) data, and in vivo transmitted light microscopy (*Figure 2*). While internal fan-fibre networks underneath the outer regions of the suction disc have been mentioned previously (*Rietschel, 1961*), we discovered that each internal fibre leads to a single microtrichium (*Figure 2b-d*). Moreover, all the microtrichia that were fractured during sample preparation appeared to be solid (in-filled) cuticular structures (*Figure 2e*). The small internal fibres leading into the microtrichia branch out from

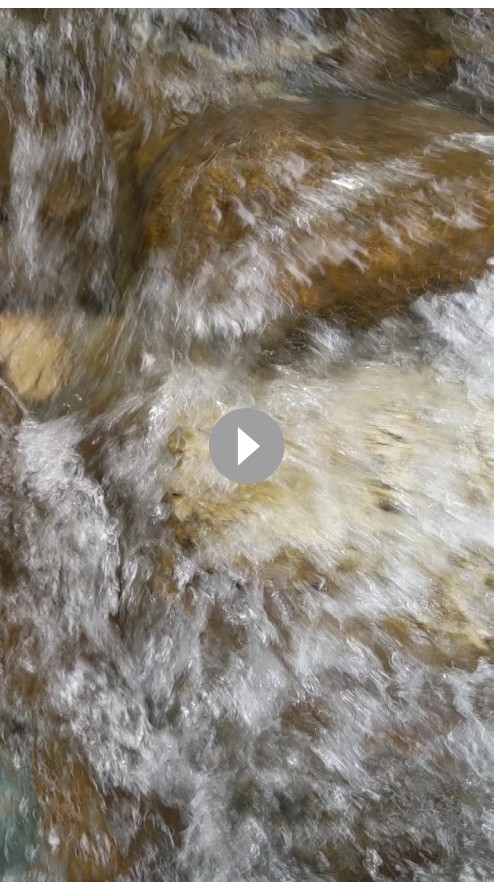

**Video 1.** Hapalothrix lugubris larvae live on rocks in torrential alpine streams. Temporarily diverting the flow of water reveals two larvae firmly attached to the rock.
https://elifesciences.org/articles/63250/figures#video1

thicker trunks originating from the ventral side of the outer radial beams (*Figure 2b-e*). Another notable ultrastructural feature is the radial beams, which are solid cuticular structures that alternate between a wide and a narrow beam (*Figure 2c*). The beams originate from the palisades, a radial zone consisting of dorsoventral cuticular rods (*Figure 2f & g*). There are 72 radial beams in a 90° segment of the disc, corresponding to 288 beams per disc (assuming no interruption from the V-notch) and a centre-to-centre spacing of around 4 μm or 1.3°.

## Attachment performance of blepharicerid larvae on different substrates

### Effect of surface roughness on the attachment performance of blepharicerid larvae

*H. lugubris* attachment forces were measured using a centrifuge force tester on smooth, micro-rough, and coarse-rough surfaces (surface profiles shown in *Table 1*). Each specimen was wetted with a droplet of water prior to centrifugation (see 'Materials and methods' section for details). Interference reflection microscopy (IRM, see below) observations showed that the contact of the suction organs under these conditions was completely wet, and no air bubbles were present in the contact zone. Peak shear and normal (adhesive) forces per body weight were measured on horizontal and vertical substrates, respectively (*Figure 3a & b*). The test substrate had a significant effect on the peak shear force per body weight, with the larvae attaching best on smooth, followed by micro-rough, then coarse-rough substrates (Kruskal-Wallis rank sum test, $\chi^2_2$ = 26.3, p<0.001; p<0.05 for all pair-wise comparisons using Dunn's post hoc tests with Bonferroni-Holm corrections). The same effect was observed for the peak normal force per body weight (Kruskal-Wallis rank sum test, $\chi^2_2$ = 24.2, p<0.001; p<0.05 for all pair-wise comparisons, see above).

## Attachment performance of three blepharicerid species on smooth surfaces

The peak shear force per body weight on smooth surfaces measured for larvae from three blephericerid species—*L. cordata*, *L. cinerascens*, and *H. lugubris*—was 585 ± 330, 324 ± 153, and 1120 ± 282, respectively (mean ± SD; *Figure 4a*). The highest overall shear force per body weight (1430) was obtained from a *H. lugubris* larva, while *L. cordata* produced a highest overall shear force of 54 mN.

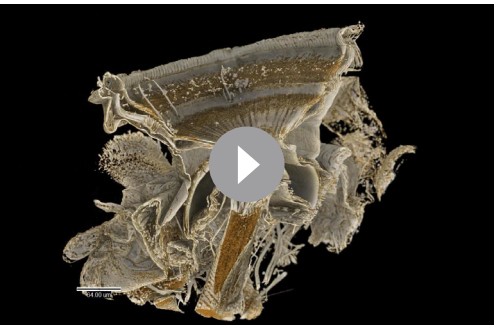

**Video 2.** Three-dimensional rendering of a *Hapalothrix lugubris* suction organ based on computed microtomography data. The video begins with a side view of the organ and its internal structures (see Figure 1c-e). Digital dissections and rendering were made using Drishti (*Limaye and Stock, 2012*).
https://elifesciences.org/articles/63250/figures#video2

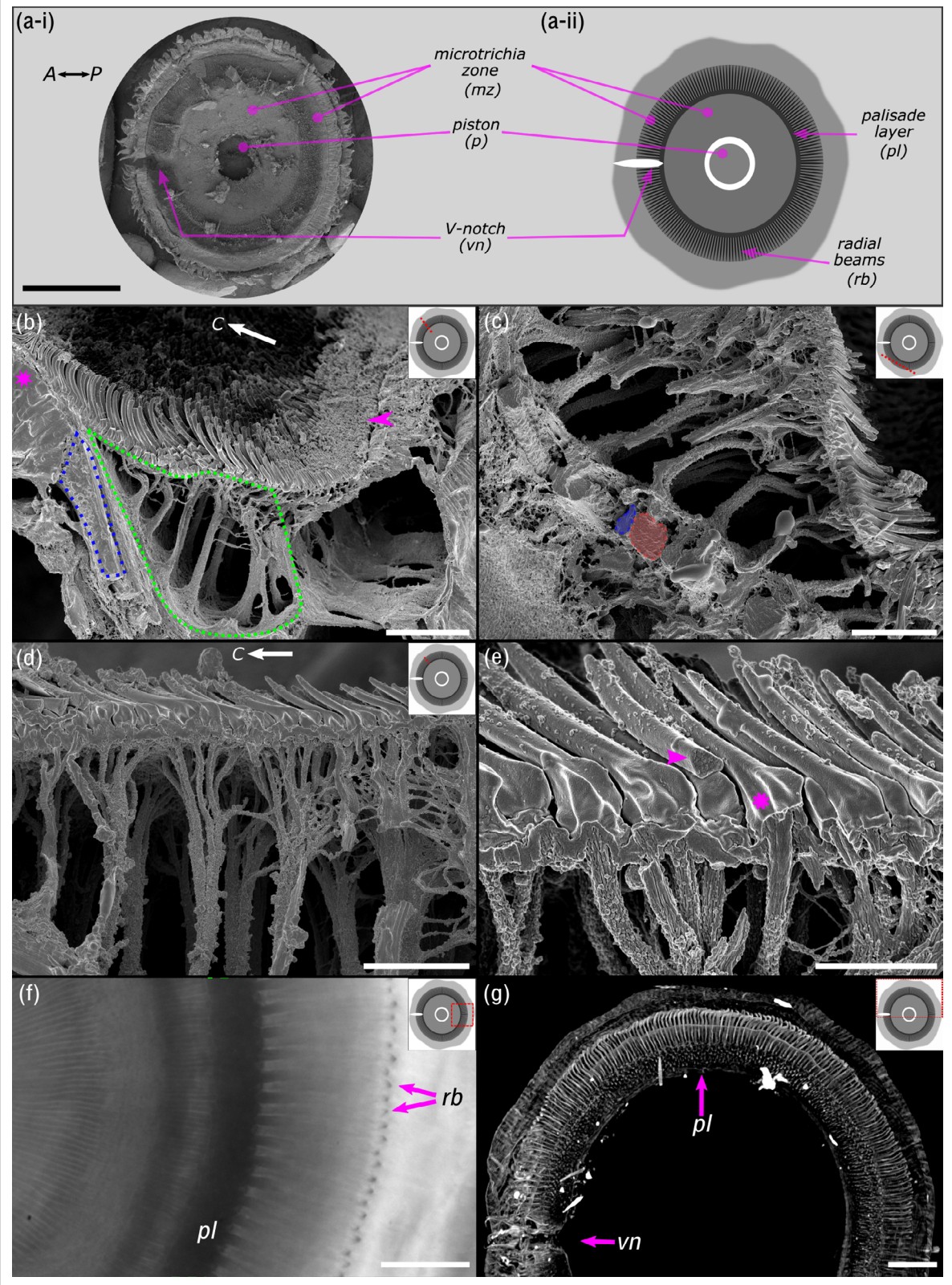

**Figure 2.** Ultrastructure of the suction disc. (**a-i**) Scanning electron micrograph showing ventral view of the suction disc. The piston is withdrawn into the suction chamber (see *Figure 1d*). (**a-ii**) Schematic of the suction disc used for subsequent panels. Note the radial beams (*rb*) are beneath the outer cuticle layer and not visible in (**a-i**) but shown in (**b**) and (**c**). (**b**) Freeze-fractured suction disc (radial fracture plane; see dotted red line on suction disc schematic). The sealing rim and its short rim microtrichia are marked by an arrowhead. Internal radial beams (encircled in blue) originate

*Figure 2 continued on next page*

*Figure 2 continued*

from the palisade layer (magenta*). The fan-fibre space is encircled in green. *C*: disc centre. (**c**) Fan-fibres extend to the radial beams, which alternate between thin (blue) and wide beams (red). (**d**) Each microtrichium connects to an internal fibre; these fibres represent the ends of thicker branched fibres originating from the radial beams. Note: spine-like microtrichia point towards the disc centre (*C*). (**e**) Microtrichia are largely solid cuticular structures (arrowhead), each connected to a fan-fibre (*). (**f**) In vivo light microscopy shows the radial beams and the palisade layer (*pl*). (**g**) Computed microtomography (micro-CT) also shows that radial beams originate from the dorsoventral palisade layer. Centre-to-centre spacing of the beams is around 4 μm or 1.3°. vn: V-notch. Scale bars: (**a**) 200 μm; (**b**) 10 μm; (**c**) 6 μm; (**d**) 5 μm; (**e**) 2 μm; (**f**) 20 μm; (**g**) 40 μm.

## Estimates of peak shear stress on smooth surfaces

Suction disc areas measured for *L. cordata* and *H. lugubris* were used to estimate the peak shear stress for blepharicerid suction attachments. Shear stresses were 41.2 ± 21.4 kPa and 39.3 ± 10.6 kPa (mean ± SD) for *L. cordata* and *H. lugubris*, respectively (*Table 2*). These values, however, are conservative estimates because (1) the contact area measurements included the outer fringe, which lies outside the suction disc seal and (2) we assumed that all six suction organs were in contact immediately before detachment. We thus derived more realistic estimates of the shear stresses by first correcting for the fact that the outer fringe layer amounted to 33 % of the total imaged contact area (n = 18 suction discs from six individuals), and second, as larvae attach with fewer suction organs prior to detachment (*Frutiger, 2002*), we assumed that three organs were in contact. Factoring in these assumptions, the shear stresses were 111 ± 57.5 kPa and 117 ± 31.4 kPa (mean ± SD) for *L. cordata* and *H. lugubris*, respectively, and the normal stresses were 120.2 ± 81.9 kPa and 71.2 ± 22.2 kPa (*Table 2*).

## Shear attachment performance of suction organs on rough substrates compared to smooth adhesive pads

While blepharicerid attachment performance decreased with increasing surface roughness, we observed a different pattern with stick insects (*Carausius morosu*), which are a model terrestrial climbing insect (*Figure 5*). Stick insects rely on a combination of smooth adhesive pads and claws for attachment, where the former facilitates strong adhesion on smooth surfaces and the latter on coarse-rough substrates. Accordingly, we found that stick insects attached equally well to smooth and coarse-rough surfaces (one-way analysis of variance (ANOVA), $F_{2,27}$ = 77.0, p = 0.97 using Tukey's post hoc test; all tests with stick insects were conducted on dry substrates). On micro-rough surfaces, however, where neither the smooth pad nor the claws proved effective, their shear force per body weight decreased 16-fold (based on the mean of back-transformed values) compared to the smooth surface (same ANOVA as above, p<0.001 using Tukey's post hoc test). The attachment performance of *H. lugubris* larvae was also affected by micro-roughness, but to a much lesser degree than in stick insects, with a twofold decrease (based on the mean of back-transformed values) in shear force per

**Table 1.** Surface profilometry of test substrates used to assess attachment performance.

| | Surface characteristics (mean ± SD) | | |
|---|---|---|---|
| Test surfaces | $R_a$ (μm) | $R_q$ (μm) | PV (μm) |
| **Rough surfaces** | | | |
| Micro-rough (0.05 μm grain size) | 0.32 ± 0.01 | 0.40 ± 0.01 | 4.56 ± 0.22 |
| Coarse-rough (30 μm grain size) | 7.97 ± 0.06 | 10.37 ± 0.08 | 78.82 ± 1.38 |
| **Microtextured substrates** | | | |
| 10 × 10 μm | NA | NA | 2.04 ± 0.18 |
| 3 × 3 μm | NA | NA | 4.48 ± 0.08 |
| 3 × 3 μm | NA | NA | 1.69 ± 0.03 |

$R_a$: average roughness (mean height deviation); $R_q$: root-mean-squared roughness; PV: maximum peak-to-valley height; NA: not applicable.

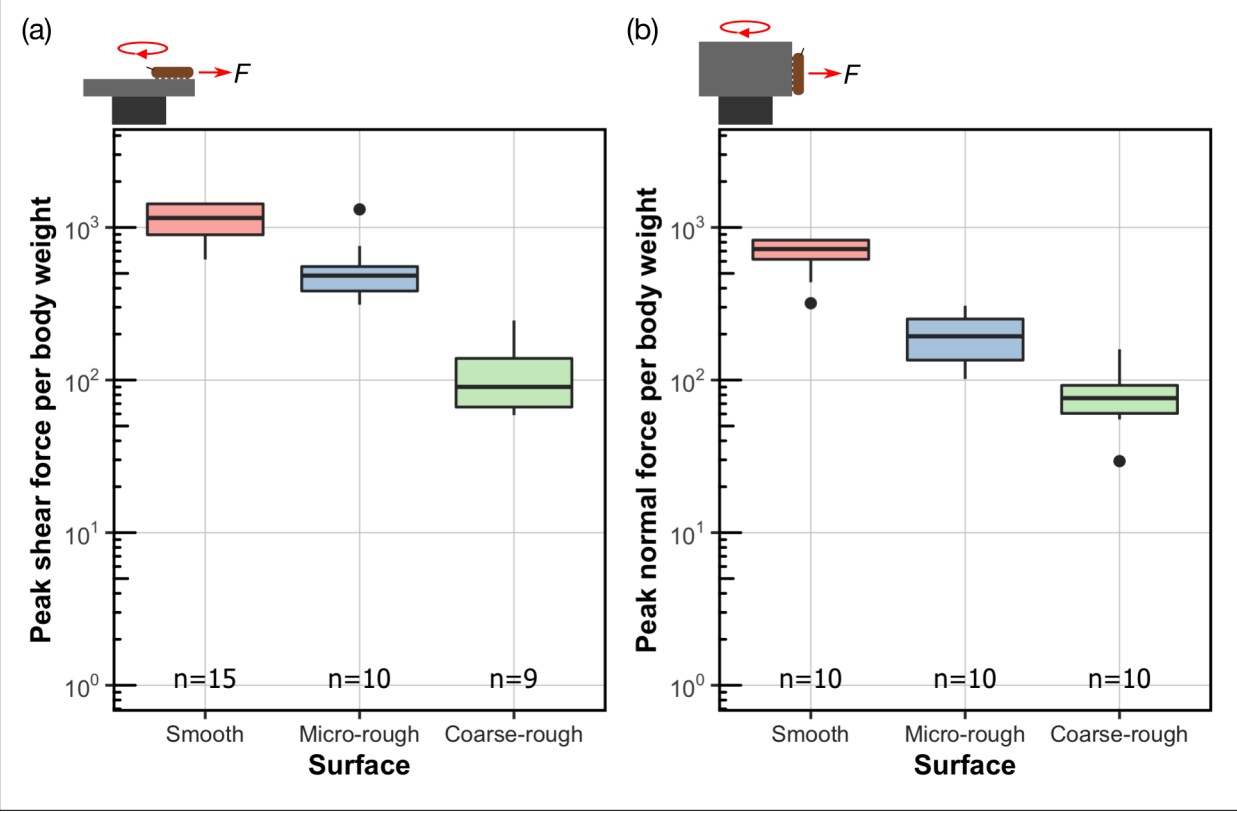

**Figure 3.** Attachment performance of *Hapalothrix lugubris* larvae on surfaces of varying roughness. *Hapalothrix lugubris* larvae performance in (**a**) peak shear force per body weight and (**b**) peak normal adhesion force per body weight. The rotation of the centrifuge is indicated by the red circular arrow. Centre lines, boxes, whiskers, and filled dots represent the median, the inter-quartile range (IQR), 1.5 times IQR, and outliers, respectively.

The online version of this article includes the following figure supplement(s) for figure 3:

**Source data 1.** Data for *Figure 3*.

body weight. It is important to mention that blephaicerid larvae do not possess any claw-like append-ages that can be used to increase grip.

## In vivo visualisation of suction organs attaching to smooth and transparent microstructured substrates

The contact behaviour of *H. lugubris* suction discs was visualised in vivo using IRM. The attachment-detachment behaviour on smooth glass resembled closely that of the related *Liponeura* species (*Kang et al., 2019*; *Frutiger, 2002*): the suction disc came into close contact with the surface at the outer fringe layer, disc rim, microtrichia zone, and around the central opening (*Figure 6ai, iii* and *Video 3*). When the piston was raised away from the surface, the greater portion of the suction disc came into close contact as a result of the reduced hydrostatic pressure. When the organ was attached, the micro-trichia made tip contact with the surface. No side contact was observed even when the suction disc was pulled closer to the surface as a result of the piston being raised. Detachment of the suction organ and forward movement was often preceded by an active opening of the V-notch (*Video 3*).

Although the outer fringe layer and the microtrichia made close contact on smooth substrates, the outcomes were different on transparent microstructured substrates made of epoxy (*Zhou et al., 2014*). On the 10 × 10 × 2 µm and 3 × 3 × 2 µm substrates (ridge width × groove width × ridge height), the microtrichia made contact on both the ridges and the grooves, which are visible as black dots in the IRM recordings (*Figure 6a-iii, b-i – iii, d-i*). Similar to our observations on smooth glass surfaces, only the tips of the microtrichia made contact on top of the ridges and inside the grooves. Side contact of the microtrichia was not observed under IRM (see *Figure 6* for a schematic of a hypo-thetical side contact). The outer fringe layer also made contact, although not uniformly. In contrast, on

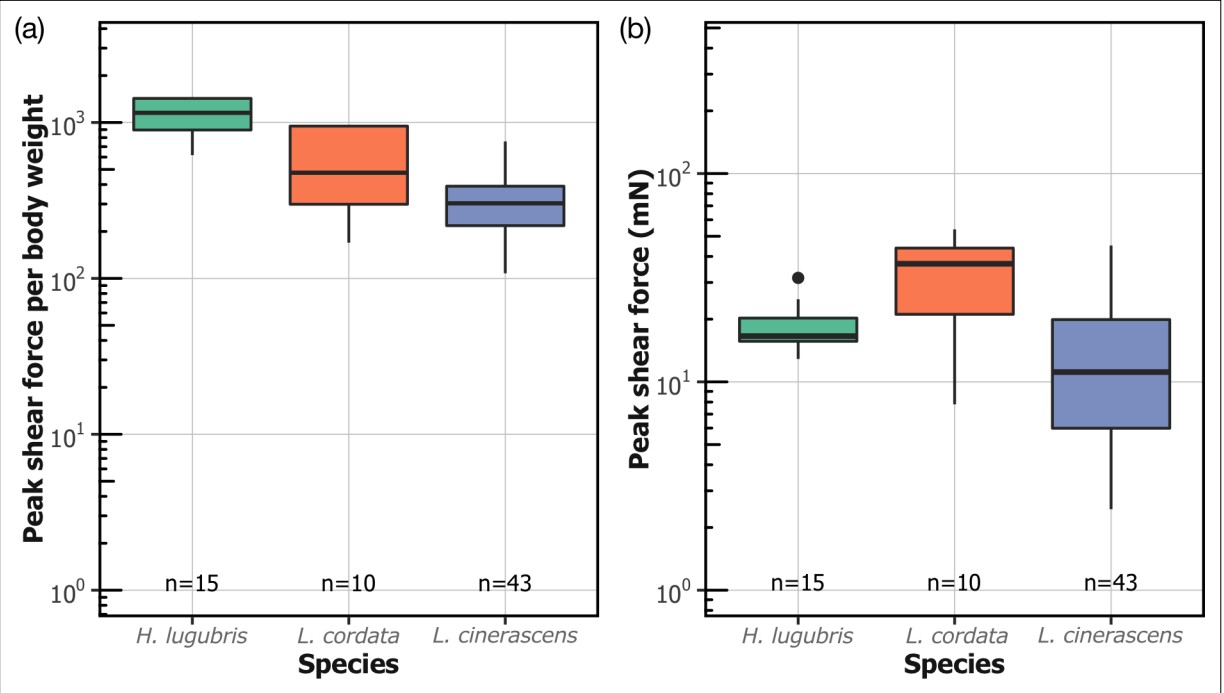

**Figure 4.** Attachment performance of three species of blepharicerid larvae (*Hapalothrix lugubris, Liponeura cordata, Liponeura cinerascens*) on smooth horizontal surface. (**a**) Peak shear force per body weight. (**b**) Peak shear force.

The online version of this article includes the following figure supplement(s) for figure 4:

**Source data 1.** Data for *Figure 4*.

the 3 × 3 × 4 µm substrate, the microtrichia and the outer fringe layer made contact only on the ridges but not inside the grooves (*Figure 6c–ii & iii*). Moreover, we observed microbial organisms freely floating and moving within the 4 -µm deep grooves (confirming the lack of close contact) but not on the ridges where the microtrichia and fringe layer were close to the surface. In contrast, no particulate or microbial movement was observed during the trials with the other microstructured surfaces (3 × 3 × 2 µm and 10 × 10 × 2 µm).

## Discussion
### Blepharicerid larvae attach with extreme strength on diverse surfaces
Blepharicerid larvae possess some of the most powerful (in terms of body weight) and complex suction organs among animals. The three species of blepharicerid larvae studied here (*H. lugubris*, *L. cordata*,

**Table 2.** Shear and normal stress estimates for suction-based attachments of *Hapalothrix lugubris* and *Liponeura cordata*.

| | Shear stress (kPa) | | | Normal stress (kPa) | | |
|---|---|---|---|---|---|---|
| **Species** | **Conservative (mean ± SD)** | **Conservative (mean ± SD)** | **n** | **Conservative (mean ± SD)** | **Realistic (mean ± SD)** | **n** |
| *Liponeura cordata* | 41.2 ± 21.4 | 111 ± 57.5 | 10 | 40.5 ± 27.6 | 120.2 ± 81.9 | 11 |
| *Hapalothrix lugubris* | 39.3 ± 10.6 | 117 ± 31.4 | 15 | 40.5 ± 27.6 | 71.2 ± 22.2 | 10 |

Conservative: contact area based on suction disc, inclusive of the outer fringe layer and all six organs in contact prior to detachment.
Realistic: based on three organs in contact immediately prior to detachment and contact areas excluding the outer fringe layer.

The online version of this article includes the following source data for table 2:

**Source data 1.** Data for *Table 2*.

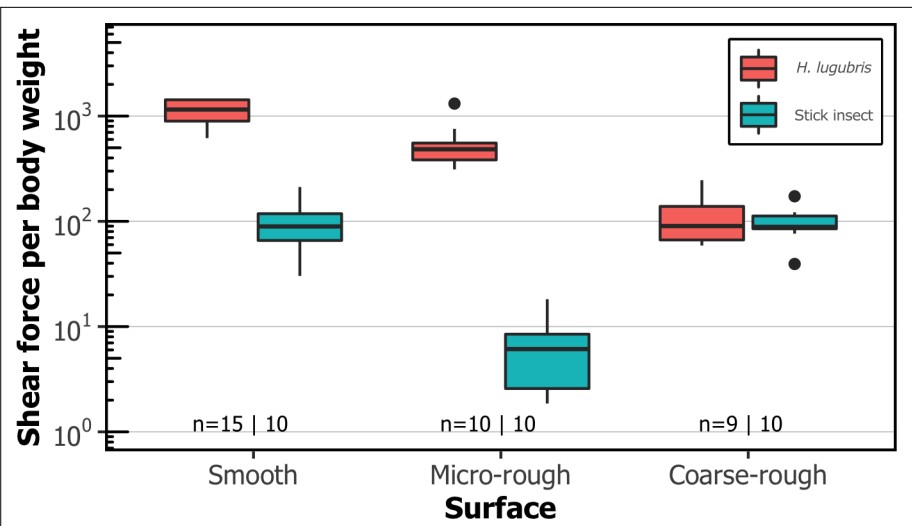

**Figure 5.** Comparison of shear attachment performance of *Hapalothrix lugubris* larvae versus stick insects (*Carausius morosus*) on smooth and rough surfaces. *Hapalothrix lugubris* larvae attach using suction organs, whereas stick insects rely on smooth adhesive pads and claws. Sample sizes are shown with *H. lugubris* on the left and stick insects on the right.

The online version of this article includes the following figure supplement(s) for figure 5:

**Source data 1.** Data for *Figure 5*.

and *L. cinerascens*) produced extreme shear forces on smooth surfaces with averages that ranged from 320 to 1120 times their own body weight. In terms of weight-specific attachment performance, the larvae performed better than all terrestrial insects measured using comparable methods (ie, whole-animal detachment experiments) (*Federle et al., 2000*; *Grohmann et al., 2014*). For example, the weight-specific shear attachment of blepharicerid larvae on smooth surfaces was 3–11 times greater than that of stick insects measured in this study. To achieve this extreme shear attachment, blepharicerid suction organs must come in close contact by generating an effective seal. Based on our in vivo visualisations of *H. lugubris* attaching to smooth glass underwater, the microtrichia make close contact with the surface, helping to both seal the organ and generate friction. This corroborates our previous findings on the suction disc contact behaviour with *L. cinerascens* and *L. cordata* (*Kang et al., 2019*). Likewise, the soft adhesive pads of stick insects make close contact on smooth surfaces, and while the weight-specific attachment forces are not as high as in blepharicerid larvae, they can withstand forces close to 100 times their body weight.

In contrast, the attachment of stick insects on micro-rough surfaces is significantly different to that of blepharicerid larvae: for stick insects, there was a 16-fold decrease in performance compared to smooth substrates, while for blepharicerid larvae, the decrease was only twofold. This difference in the impact of micro-roughness can be attributed to the two fundamentally different mechanisms of attachment: on micro-rough surfaces, neither the soft adhesive pads nor the tarsal claws of stick insects function properly (*Bullock and Federle, 2011*). This is in part due to the reduced effective contact area (the adhesive pads cannot mould sufficiently to the asperities) and also due to the reduced friction from tarsal claws (the claws cannot interlock with the small asperities). On the other hand, blepharicerid suction organs are still able to seal on micro-rough surfaces and microtrichia can interact with the asperities, which likely explains why their performance was not as diminished as in the stick insects. In addition, blepharicerid suction organs may adhere better to micro-rough surfaces because during partial contact, the gaps between the detached regions and the substrate are filled with water, whereas detached regions of stick insect pads are filled with air. As water is effectively incompressible and approximately 50 times more viscous than air, the water-filled contact zone can provide a much stronger resistance against detachment even under conditions of partial contact as on micro-rough substrates.

While blepharicerid larvae attached more strongly than stick insects on micro-rough surfaces, the opposite was found on coarse-rough surfaces: for stick insects, there was no difference in performance

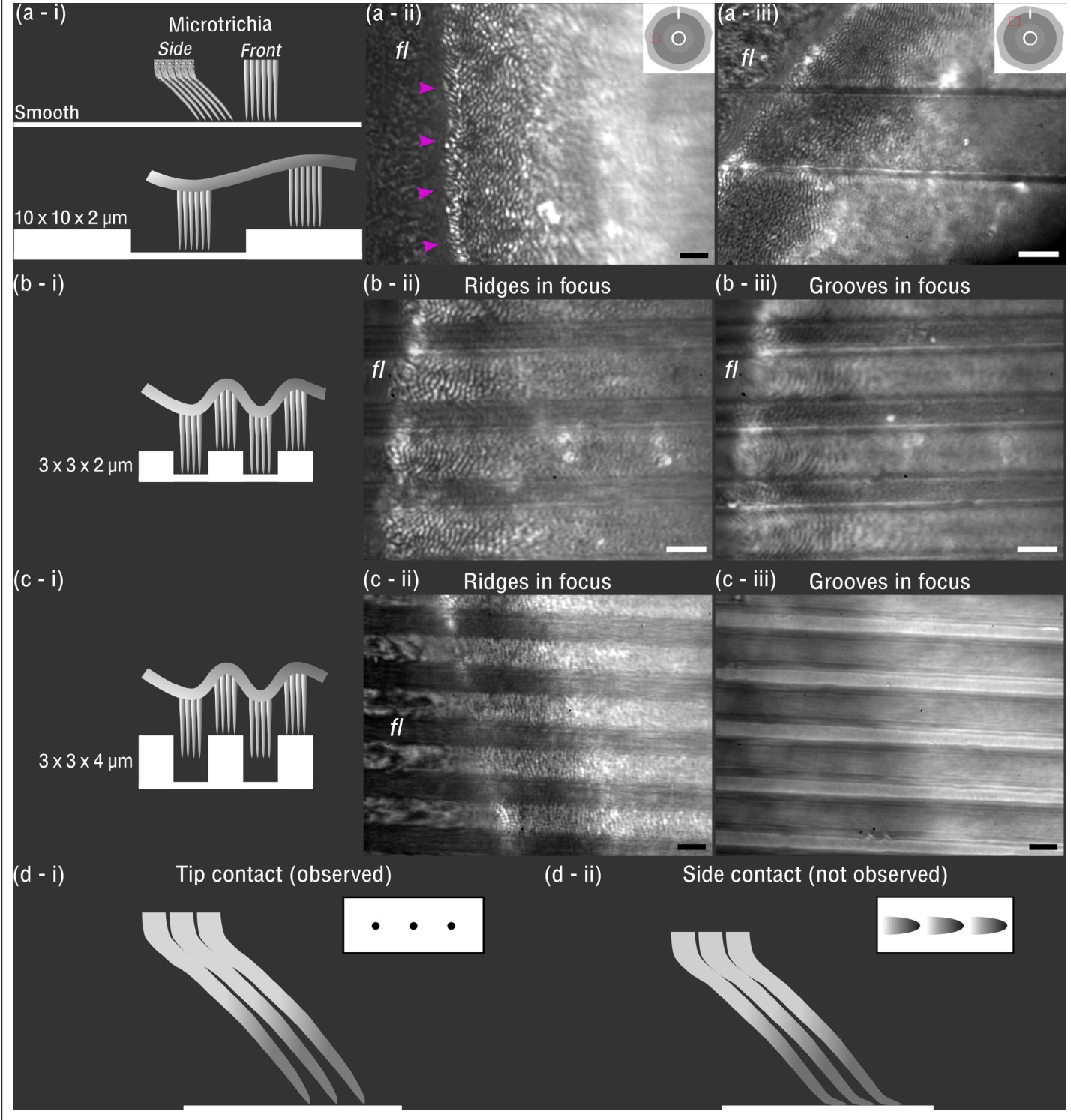

**Figure 6.** In vivo visualisation of *Hapalothrix lugubris* suction disc contact on different substrates. (**a-i**) Schematic of the microtrichia on smooth and 10 × 10 × 2 μm microstructured surface (ridges and grooves, 10 μm in width, and grooves, 2 μm deep) (shown to scale). (**a-ii**) On smooth glass, microtrichia made tip contact (seen as black dots under interference reflection microscopy [IRM]). Outer fringe layer (*fl*) is outside the seal (arrowheads). (**a-iii**) On 10 × 10 × 2 μm substrates, contact from microtrichia and *fl* was similar to the contact on the smooth surface. (**b-i and b-ii**) On 3 × 3 × 2 μm substrates, the microtrichia made tip contact on the ridges, as well as in the grooves, as seen in (**b-iii**). However, fewer microtrichia made contact within the narrow grooves compared to the 10 × 10 × 2 μm surface. Note that (**b-ii**) and (**b-iii**) differ only in the focus height. (**c-i to c-iii**) On 3 × 3 × 4 μm substrates, microtrichia made close contact on the ridges, but inside the deep grooves there was no contact. (**d-i**) Schematic of microtrichia coming into tip contact

*Figure 6 continued on next page*

*Figure 6 continued*

on smooth glass (inset: contact area observed under IRM). (**d-ii**) Schematic representing a hypothetical scenario that was not observed where the microtrichia tips bend and make side contacts. Scale bars: 3 µm for all microscopy images.

between coarse-rough and smooth surfaces, whereas blepharicerid larvae attachment decreased 11-fold. It is likely that both blepharicerid suction organs and stick insect adhesive pads are unable to cope with coarse surface roughness. The adhesive pads of both insects may be unable to fully mould to the large asperities, and the length of the microtrichia may be insufficient to reach the lower regions of the surface profile (*Figure 6*). Stick insects, however, have large pretarsal claws that can interlock with large asperities for strong attachment. Previous studies on dock beetles (*Gastrophysa viridula*) and stick insects (*C. morosus*) have reported that both beetle and stick insect attachments on coarse-rough surfaces decrease significantly when the claws are removed (*Bullock and Federle, 2011*; *Scholz et al., 2010*). This means that, although stick insects and dock beetles use two distinct adhesive systems (smooth versus hairy pads), the combination of the claws and the adhesive pads produces the same trend: both insects attach strongly to smooth and coarse-rough surfaces but poorly to micro-rough surfaces. In contrast, blepharicerid larvae do not have claw-like appendages and rely on suction organs for attachment. Consequently, these aquatic larvae do not follow the same trend as terrestrial climbing insects and perform the worst on coarse-rough substrates. A similar result was reported by *Liu et al., 2020* using *Blepharicera* sp., where the larval attachment performance decreased with increasing surface roughness, although no quantitative information on attachment forces can be extracted from their study (the study also used a centrifuge, but only reported the rotation speed but not the insects' mass and position; it is also unclear whether the larvae were wetted prior to the tests).

## Blepharicerid suction organs withstand stresses comparable to those by other biological suction organs

Although attachment forces per body weight help to assess the performance from a biological perspective, the attachment force per contact area, or the stress, is needed to make comparisons between animals that adhere to surfaces. In blepharicerid larvae, the shear stress on smooth substrates ranged from 39 to 117 kPa, and from 24 to 120 kPa for normal stress (the range in values arises from the assumptions used to calculate the contact area and the number of organs remaining in contact immediately prior to detachment). These values are similar to those reported in the literature for suction attachments of other animals on smooth substrates: the remora fish can withstand 93 kPa in shear and 38 kPa in the normal direction (*Fulcher and Motta, 2006*); octopus can resist normal stresses of up to 271 kPa, squids up to 830 kPa, and lumpsucker fish up to 102 kPa (*Smith, 1996*; *Davenport and Thorsteinsson, 1990*). Like the cephalopods, blepharicerid suction stress can surpass 101 kPa (1 atm at standard sea level and temperature), with one *L. cordata* reaching 228 kPa (realistic estimate; 77 kPa if based on conservative assumptions, as outlined in 'Results' section). Although a pressure difference of 1 atm is considered the upper threshold for suction attachments in air, this is not the case if the contact zone is completely wet and bubble-free, as the strong cohesion of water allows suction stresses to exceed 1 atm (*Smith, 1991*; *Smith, 1996*; *Wang et al., 2019*; *Wang et al., 2020*). Even if they do not reach 1 atm, blepharicerid larvae can generate sufficient attachment force to resist the fast flow rates in their natural habitat.

## Ultrastructural components may help to stabilise the suction disc under high stress

Recent work on cupped microstructures (which resemble microscopic suction cups) has revealed the mechanisms of failure in underwater suction attachments: (1) under sustained tensile stress, the rim slides inwards and the rim diameter contracts by ~30 %; (2) immediately prior to detachment,

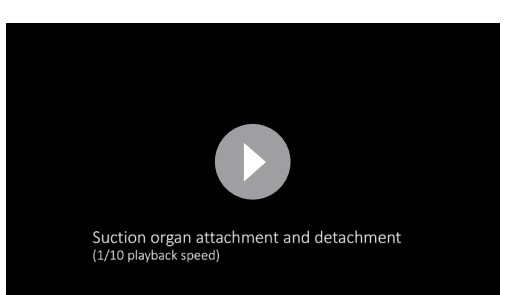

**Video 3.** Suction organ of a *Hapalothrix lugubris* larva in action, filmed using in vivo interference reflection microscopy and a custom flow chamber. Note the V-notch opens immediately prior to detachment.
https://elifesciences.org/articles/63250/figures#video3

sections of the rim buckle inwards, leading to adhesive failure (*Wang et al., 2019*). Similar failure modes were also reported for macroscopic suction cups (*Ditsche and Summers, 2019*). Two structural features of the blephaicerid suction organs could represent adaptations to counter the aforementioned failure mechanisms seen in cupped microstructures. First, the internal radial beams can provide structural support to reduce inward sliding and buckling of the suction disc rim. Similar to how the flexible membrane of an umbrella is stiffened by radial spokes, these stiff cuticular radial beams can stabilise the suction disc when the organ is under high tensile stress. Bones within the clingfish suction organ may also prevent inward sliding (*Wang et al., 2020*). While we have yet to visualise blephaicerid suction organs fail under extreme forces, their powerful attachments suggest that they possess mechanisms to counter suction cup failure.

The second morphological feature that may reduce inward sliding of the rim is the numerous microtrichia in the contact zone of the blephaicerid suction disc. The microtrichia can interlock with surface asperities and minimise inward sliding. This has been reported for the remora fish, which have stiff posterior-facing structures called spinules within their suction pads that passively engage with asperities during high-drag conditions (*Beckert et al., 2015*; *Fulcher and Motta, 2006*), as well as for the clingfish, where a hierarchical system of rods and filaments near the periphery of the suction organ may increase friction on rough surfaces (*Ditsche and Summers, 2019*). Similarly, blephaicerid microtrichia are naturally angled (~40° to 50° relative to the horizontal) and point towards the centre of the suction disc (*Kang et al., 2019*). Hence, inward sliding would passively lead to additional interlocking of the microtrichia tips with rough surfaces, thereby loading the microtrichia along their axis.

To interlock effectively, structures like the remora spinules need to be stiff and strong (*Dai et al., 2002*; *Wang et al., 2017*). The blephaicerid microtrichia are indeed likely to be stiff structures since (1) they are made of solid cuticles (a composite material of chitin fibres embedded in a matrix of proteins), and the dense, sclerotised cuticle can reach high elastic moduli [*Parle et al., 2017*]; (2) we never observed any microtrichia in side contacts, even on micro-structured surfaces when the disc was pressed into contact. From the observation that microtrichia only make tip contact, and assuming that they are loaded equally, we can give a conservative lower estimate for the elastic modulus of the microtrichia cuticle by following *Goss and Chaouki, 2016* and modelling the microtrichium as a cylindrical beam loaded at an angle (*Figure 7*; see 'Materials and methods' section for details). Based on this model, the microtrichia cuticle must have a stiffness of at least 0.3–0.4 GPa, similar to the stiffness of wood and bone (*Vincent and Wegst, 2004*), to prevent any side contact (for an angle to the disc surface of 40° to 50°; see *Figure 7*). It is not unlikely that the microtrichia cuticle is even

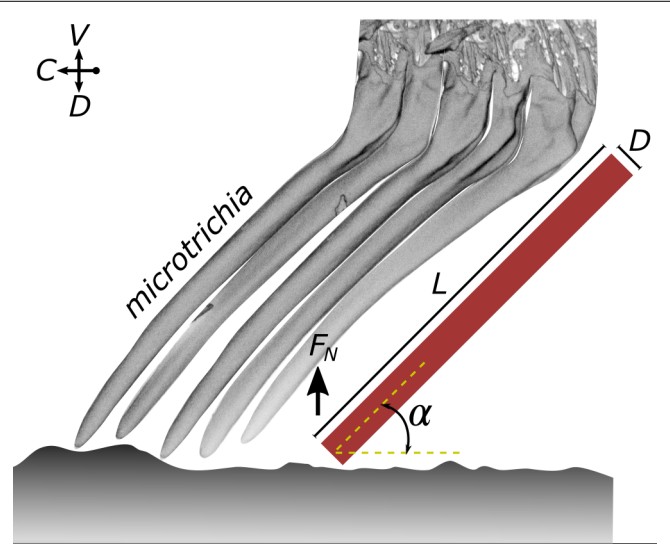

**Figure 7.** Estimate of the elastic modulus of the microtrichia cuticle needed to maintain tip contact during attachment. A microtrichium is modelled as a cylinder (length $L$ and diameter $D$: 6.7 μm and 0.56 μm, respectively) that is loaded with a peak normal force ($F_N$) and that makes tip contact at an angle (40° to 50°). See text for further details on the assumptions and the model.

stiffer: sclerotised cuticles can have an elastic modulus of up to 20 GPa (*Vincent and Wegst, 2004*; *Wegst and Ashby, 2004*; *Sykes et al., 2019*). This supports our idea that microtrichia can maintain tip contact during interactions with rough surfaces and could serve a similar function to stiff remora spinules.

It is worth mentioning that, while stiff interlocking structures play an important role in increasing friction on rough surfaces, an effective seal is crucial for attachments on both smooth and rough surfaces. In remora fish and clingfish, a soft rim helps the organ to mould to the surface and form a seal (*Wang et al., 2020*). In bio-inspired suction devices as well, it is the combination of stiff structures and a soft sealing rim that generates the strongest attachments to smooth and rough surfaces (*Ditsche and Summers, 2019*; *Wang et al., 2017*). The detailed biomechanics of how the dense array of microtrichia produces a tight seal on rough surfaces and interlocks with small substrate asperities is beyond the scope of this study and remains to be explored in future work.

## Blephaof suction organs are unique among insects and specialised for fast-flow conditions

We have demonstrated that blephaand suction organs can attach with extreme strength to both smooth and rough surfaces. Despite the potential strength of each attachment, the larvae are surprisingly mobile in their natural habitat (*Frutiger, 1998*; *Federle and Labonte, 2019*). Both attachment force and mobility are required for blephaand larvae to survive in their challenging habitats, which include raging alpine torrents and areas near the base of waterfalls. Currently, blephaand suction organs are the only examples of piston-driven suction organs in insects. While the circular setae of male diving beetles (Dytiscidae) are also considered to be suction organs, the two systems have markedly different morphologies: (1) the ventral surfaces of circular setae are comparatively smooth; (2) circular setae lack a muscle-driven central piston; (3) there are no known mechanisms for rapid detachment in dytiscid circular setae (*Chen et al., 2014*; *Karlsson Green et al., 2013*; *Nachtigall, 1974*). As male dytiscid beetles use their suction organs to attach to the smooth sections of the female's pronotum and elytra, their attachment works best on smooth surfaces and their performance declines strongly on rough surfaces (*Karlsson Green et al., 2013*). Researchers have suggested that male and female dytiscid beetles are engaged in an evolutionary arms race driven by sexual conflict, as female cuticular surfaces are modified to hinder male attachment with suction discs (*Karlsson Green et al., 2013*; *Bergsten et al., 2001*). Interestingly, it appears that male beetles did not evolve friction-enhancing structures on their circular setae to facilitate adhesion to rougher regions of the female cuticle. In contrast, blephaand suction discs are densely covered in microtrichia that likely enhance the grip on rough surfaces in high-drag conditions. This difference in morphology may be based on function or phylogenetic constraints: male diving beetles may not need to attach to rough elytra if their setae can generate sufficient attachment forces on smooth cuticle alone; alternatively, beetle setae may be more limited in the structures that can be developed from them, compared to the blephaand organ, which is highly complex and multicellular.

A more comparable suction-based attachment system can be found in remora fish. Remora fish use suction pads—highly modified dorsal fin spines—to attach to sharks, whales, and manta rays (*Beckert et al., 2015*; *Fulcher and Motta, 2006*). Recent studies on the functional morphology of remora suction pads have greatly expanded our understanding on the mechanisms underlying their impressive performance (*Beckert et al., 2015*; *Wang et al., 2017*; *Gamel et al., 2019*). A remora suction pad comprises a soft fleshy outer rim and rows of lamellae topped with spinules. The pitch of the lamellae is muscle-controlled to facilitate spinule contact with rough surfaces, such as shark skin. When engaged, the tips of these stiff spinules interlock with surface asperities and increase friction, thereby increasing shear resistance. Moreover, the angled posterior-facing lamellae and spinules promote passive engagement when subjected to shear forces from a swimming host (*Beckert et al., 2015*; *Fulcher and Motta, 2006*). The similarities between the remora suction pad and the blephaand suction organ may be based on overlapping functional requirements, as both animals have to cope with high shear forces (fast-swimming hosts for the remora and torrential rivers for blephaand larvae).

## Effect of the biofilm layer on suction attachments to natural rock surfaces

Since blepharicerid larvae attach to rocks underwater and feed on epilithic algae, their suction organs will in most cases contact the biofilm, yet the details of this interaction are unknown (*Frutiger and Buergisser, 2002*). As hypothesised previously, it is possible that the stiff microtrichia penetrate the biofilm layer (*Rietschel, 1961*; *Nachtigall, 1974*). This may allow the microtrichia to directly interlock with asperities on the rock surface or to generate additional friction from embedding numerous microtrichia into the biofilm. Mayfly larvae, which also inhabit fast-flowing watercourses, have friction-enhancing hairs that benefit from interacting with the biofilm (*Ditsche et al., 2014a*). It was found that a higher proportion of mayfly larvae can withstand fast flow rates on smooth hard (epoxy) substrates when the biofilm is present. Moreover, the setae and spine-like acanthae on the ventral surfaces of mayfly larvae can generate friction forces on clean rough substrates (*Ditsche-Kuru et al., 2010*). Additional experiments with blepharicerid larvae are underway to investigate the interaction between stiff microtrichia and soft substrates.

To conclude, we have shown that blepharicerid larvae use their suction organs to generate extreme attachment to diverse surfaces. The suction organ morphology is conserved between *Hapalothrix* and *Liponeura* larvae, and consists of a suction disc that contacts the substrate, dense arrays of microtrichia on the disc surface, muscles to control the piston, and the V-notch detachment system. We characterised the suction disc ultrastructure, which includes internal radial beam structures that could help to stabilise the suction disc when subjected to high stress, and fan-fibres that connect individual microtrichia to the radial beams. In terms of attachment performance, blepharicerid larvae withstand extreme shear forces equivalent to 320–1120 times their body weight on smooth substrates, depending on the species. *H. lugubris* performed the best overall, reaching shear forces equivalent to 1430 times the body weight, as well as an estimated shear stress of 117 kPa and normal stress of 24 kPa. Although their attachment decreased with increasing surface roughness, blepharicerid suction organs performed better than the smooth adhesive pads of stick insects on micro-rough surfaces. We confirmed that blepharicerid suction organs can mould to large surface asperities and that microtrichia come into close contact between the asperities. These microtrichia are stiff spine-like structures that are specialised for maintaining tip contact with the surface for interlocking with asperities. Our study provides new insights into the function of a highly adapted insect adhesive organ and expands our understanding of the function of biological suction organs.

## Materials and methods

### Sample collection and maintenance

*L. cinerascens* (Loew, 1845) larvae were collected from fast-flowing alpine rivers near Meiringen, Switzerland (GPS location 46° 44' 05.6" N, 8° 06' 55.4" E, in May 2018), and close to Grinzens, Tirol, Austria (47° 12' 41.4" N, 11° 15' 28.1" E, in September 2018). At the latter site, *L. cordata* (Vimmer, 1916) and *H. lugubris* (Loew, 1876) were also collected. For all the species, we collected third and fourth instar larvae that were large enough to be handled for experiments. Wearing fishing waders and diving gloves, we removed rocks from the most turbulent areas of the river and brought them to the riverbank for specimen collection. Although it was previously noted that the larvae can attach so firmly that they are torn upon detachment (*Komárek, 1914*), we found that a gentle nudge using soft-touch tweezers can elicit an evasive response from them, whereupon they could be easily picked up using tweezers and placed in specially prepared 50 ml Falcon tubes. All larvae were kept in these tubes in an ice box during collection and transport. Rocks were returned to their approximate locations after collection.

For long-term maintenance of the larvae, an aquarium tank was set up with water and small rocks from the collection site. A filter unit with two outlets for a small water cascade was used to filter the water and to simulate the natural environment. Multiple air pumps were also placed close to the aquarium walls to provide ample oxygenation and additional regions with turbulent flow. To promote algal growth, an over-tank light-emitting diode (LED) light was set to a 12 -hr day-night cycle. The aquarium was kept in a 4 °C climate room (mean temperature of 3.2°C ± 0.9°C; mean ± SD) to replicate alpine stream temperatures.

## Scanning electron microscopy of *Hapalothrix lugubris* suction organs

SEM was used to image fourth instar *H. lugubris* larvae as described previously (*Kang et al., 2019*). In brief, samples fixed in 70 % ethanol (v/v) were flash-frozen in liquid ethane cooled with liquid nitrogen and freeze-fractured immediately afterwards with a double-edged razor blade on a cooled aluminium block to obtain longitudinal views. Samples were freeze-dried overnight, then carefully mounted on SEM aluminium stubs using carbon tape and silver paint. They were then sputter-coated with 15 nm of iridium and imaged using a field-emission SEM (FEI Verios 460).

## X-ray microtomography of blephaticerid suction organs

One *H. lugubris* fourth instar larva was fixed in 2 % paraformaldehyde and 2 % glutaraldehyde (v/v) in 0.05 M sodium cacodylate buffer (pH 7.4) for 7 days at 4 °C. The larva was then dissected into six pieces—each containing one suction organ—and fixed for an additional day. The samples were then rinsed multiple times in 0.05 M sodium cacodylate buffer followed by deionised water before dehydration through a graded ethanol series: 50%, 75%, 95%, 100 % (v/v), and 100 % dry ethanol. The dehydrated samples were critical-point dried using four flushes of liquid $CO_2$ in a Quorum E3100.

One critical-point-dried suction organ was used for imaging via micro-CT. The sample was mounted on a standard dressmaker's pin using ultraviolet (UV)-curable glue, then imaged using a lab-based Zeiss Xradia Versa 520 (Carl Zeiss XRM, Pleasanton, CA, USA) x-ray microscope. The sample was scanned at 0.325 µm/pixel with an accelerating X-ray tube voltage of 50 kV and a tube current of 90 µA. A total of 2401 projections collected at 20 s exposure intervals were used to perform reconstruction using a Zeiss commercial software package (XMReconstructor, Carl Zeiss), utilising a cone-beam reconstruction algorithm based on filtered back-projection. Subsequent 3D volume rendering and segmentations were carried out using Dragonfly v4.0 (Object Research Systems Inc, Montreal, Canada) and Drishti v2.6.5 and v2.7 (*Limaye and Stock, 2012*).

## Measuring attachment performance of blephaticerid larvae using a centrifuge

Insect attachment forces were measured using a custom centrifuge set up described previously (*Federle et al., 2000*). The centrifuge operated on the following principle: a platform with the test substrates and the insect was driven by a brushless motor, and a light barrier sensor was triggered per rotation. This signal was used to synchronise image acquisition from a USB camera (DMK 23UP1300; The Imaging Source Europe GmbH, Bremen, Germany), and image frames and their corresponding times were recorded using the StreamPix4 software (NorPix Inc, Montreal, Canada). For safety reasons, the maximum centrifugation speed was limited to approximately 75 rotations per second (rps). Some of the blephaticerid larvae could not be detached even at the maximum speed (n = 14 out of 136 measurements); in such cases, we used the maximum acceleration of a successfully detached individual from the given species.

Effect of surface roughness on the peak shear force of blephaticerid larvae was measured on the following substrates: smooth (clean polyester films), micro-rough (polishing films with a nominal asperity size of 0.05 µm; Ultra Tec, CA, US), and coarse-rough (30 µm polishing films; Ultra Tec). The same substrate types were used to measure the normal force (substrates mounted vertically in the centrifuge), but a polished polymethyl methacrylate (PMMA) surface was used as the smooth substrate. Surface characteristics (average roughness (mean height deviation) $R_a$, root-mean-squared roughness $R_q$, and maximum peak-to-valley height (PV)) of the micro-rough substrates were obtained using white-light interferometry with a scan area of 0.14 × 0.10 mm (Zygo NewView 200; Zygo Corporation, CT, USA). Micro-rough substrates were sputter-coated with 5 nm of iridium prior to scanning to improve the surface reflectivity. As the coarse-rough substrate could not be adequately imaged via white-light interferometry, we used a Z-stack image focal-depth analysis technique as described elsewhere (*Sarmiento-Ponce et al., 2018*) with a scan area of 0.44 × 0.58 mm. For both surfaces, three regions were selected at random and imaged. Interferometry images were analysed using MetroPro software (Zygo), and a custom MATLAB script was used to reconstruct the surface profile from the Z-stack images (The MathWorks Inc, MA, United States).

Since *L. cinerascens* and *L. cordata* were difficult to maintain in laboratory conditions, these two species were tested only on smooth horizontal surfaces (n = 43 and n = 10, respectively). The full range of tests on smooth, micro-rough, and coarse-rough surfaces was conducted for *H. lugubris* (n =

9–15 for shear tests; n = 10 for all tests in normal direction). Prior to the experiments, individuals were selected from the laboratory aquarium and placed inside specially prepared 50 ml Falcon tubes. This tube was kept on ice for the duration of the experiment. For each run, a larva was carefully removed from the tube and placed on the test surface. A droplet of water (taken from the aquarium) was used to wash excess debris from the insect, and lab tissue paper was used to wick away excess water without removing all moisture from the larva; the contact zone of the suction discs was still completely wetted under these conditions, as confirmed by IRM observations. These steps were necessary to prime the larvae for the centrifugation trials as they often displayed defensive behaviour while being handled. The larvae adhered and remained still once primed, and between two and four repetitions were performed for each larva. All centrifuge trials were conducted within 7 days of collection.

After the trials, all the larvae were blot-dried on filter paper and weighed using an analytical balance (1712 MP8; Sartorius GmbH, Göttingen, Germany). Statistical analyses were conducted on $log_{10}$-transformed values using R v3.6.2 run in RStudio v 1.2.5033 (*R Development Core Team, 2019*; *Team RS, 2019*).

### Calculating peak stress values on smooth horizontal plastic surfaces

Normal (adhesive) stress and shear stress (defined as the peak attachment force divided by the contact area) were calculated using suction disc areas measured for *L. cordata* and *H. lugubris* larvae. Larvae were placed on microscope slides so that the suction organs fully contacted the glass and imaged with a stereomicroscope. Every tested *L. cordata* and *H. lugubris* specimen was imaged. A representative organ was selected from each *L. cordata* and *H. lugubris* larva, and the contact area calculated by fitting a circle inclusive of the outer fringe layer using FIJI (*Schindelin et al., 2012*) (https://imagej.net/Fiji). The peak attachment force was then divided by this contact area to determine the peak stress for each larva.

### Measuring peak shear and normal attachment forces of stick insects

*C. morosus* (Sinéty, 1901) stick insects were used as a model for terrestrial insect adhesion, and their attachment on surfaces with varying roughness was measured to compare against blepharicerid larval

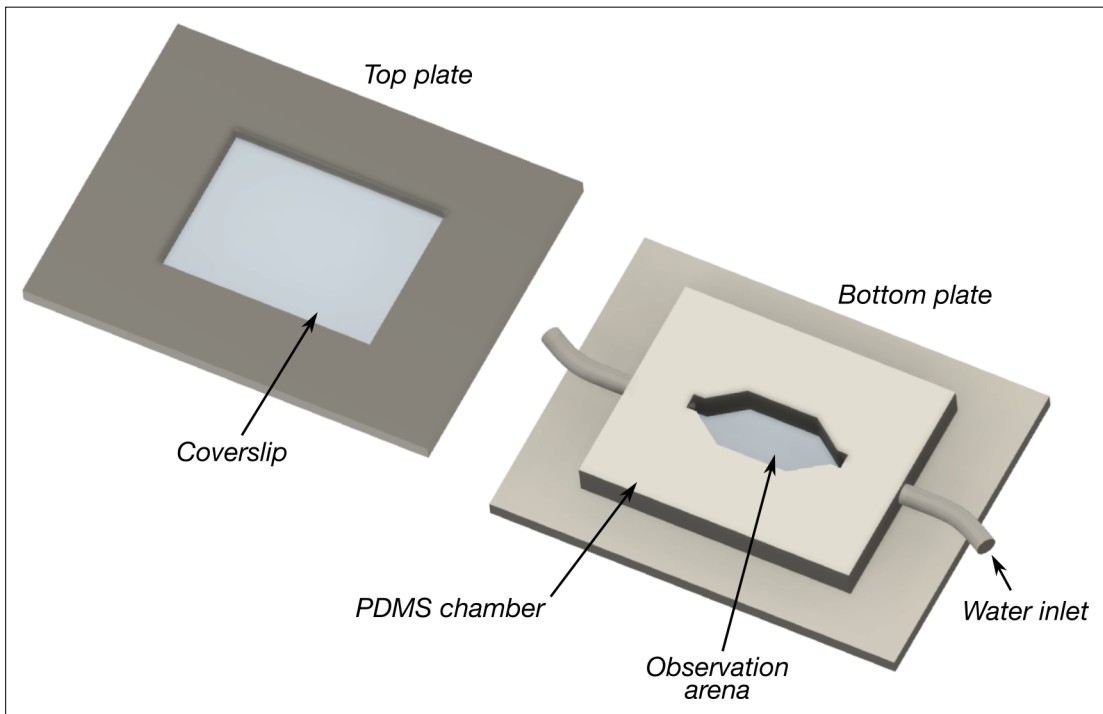

**Figure 8.** Schematic of the flow chamber used to observe blepharicerid larvae locomoting in fast-flow conditions. Two to five larvae at a time were added to the observation arena, covered with the top and bottom coverslips and plates, and imaged using interference reflection microscopy. A water pump continuously circulated cooled water at flow rates ranging between approximately 6 and 15 ml/s. Coverslip thickness: 0.16–0.19 mm. Plate dimensions: 100 × 60 mm in length × width. Observation arena dimensions: approximately 16 × 8 mm in width × height. PDMS: polydimethylsiloxane.

attachment. Second instar nymphs with undamaged legs and tarsi were selected for centrifuge experiments using smooth, micro-rough, and coarse-rough surfaces (n = 10 per surface). No normal forces could be measured on the micro-rough surface as stick insects failed to hold their body weight during preliminary tests. Before each trial, we checked that the specimen was in contact with the surface using all six legs and that the surface was uncontaminated. Stick insects were oriented with the head facing out, and each individual was tested twice and weighed afterwards. The higher attachment force per individual was used as the peak attachment force.

## In vivo observation of suction organs attaching to smooth and micro-patterned substrates

In order to examine blepharicerid larvae locomoting for extended periods of time, a custom flow chamber was built to imitate the fast-flow conditions of their natural environments (*Figure 8*). Two aluminium plates (approximately 60 × 100 mm in height × width), each with a rectangular window, were used to sandwich an inner chamber made out of polydimethylsiloxane (PDMS; Sylgard 184, Dow Corning, MI, USA). This inner chamber had a lemon-shaped chamber to serve as the observation arena, and an inlet and an outlet for water circulation. Two microscope coverslips (0.16–0.19 mm thickness; Agar Scientific, Stansted, UK) were used to encase the inner chamber. Two to five larvae were placed on the bottom coverslip, and once the top coverslip was placed over the arena, four clamps were used to squeeze the aluminium plates and coverslips against the PDMS. The soft PDMS moulded closely to the plates and created a water-tight seal. Aquarium water (kept cool in an ice bath) was pumped via a micro-pump (M200S-V; TCS Micropumps Ltd, UK), and the input voltage was controlled by a microprocessor. The flow rate was controlled by setting an appropriate pump-operating voltage. With this flow chamber, we recorded *H. lugubris* larvae locomotion and the attachment/detachment of suction organs on smooth glass surfaces via IRM, which has been previously used to investigate the contact between animal adhesive organs and the substrate (*Federle et al., 2006*; *Federle et al., 2002*). Videos were recorded using a USB camera (DMK 23UP1300) and the IC Capture software (v2.4.642.2631; The Imaging Source GmbH) at 30 frames per second (FPS).

To observe how suction organs respond to surface roughness, we used transparent micro-structured surfaces with well-defined micro-ridges and grooves fabricated by photolithography and nanoimprinting (*Zhou et al., 2014*). In brief, a master surface was first produced using photolithography, and a PDMS mould of this master was used to cast the final surface out of epoxy. Three micro-ridge geometries were used in our experiments: (1) 3 × 3 × 2 µm (ridge width × groove width × ridge height); (2) 3 × 3 × 4 µm; and (3) 10 × 10 × 2 µm. As the ridge height is only approximately controlled through the spin-coating of photoresist when producing the master, we measured it from the epoxy replicas using white-light interferometry as mentioned above (see *Table 1*). Four to five regions from each uncoated substrate were imaged and only regions without artefacts were used in the final calculation. For simplicity, when referring to the substrates, the depths of the grooves were reported to one significant figure (ie, 3 × 3 × 2 µm, 3 × 3 × 4 µm, and 10 × 10 × 2 µm, for widths of ridges, grooves, and ridge height). Note that as these surfaces could not be used in combination with the flow chamber, a *H. lugubris* larva was placed on the substrate, wetted with a droplet of aquarium water, gently motivated with soft-touch forceps, and recorded as they moved around on the substrate. One *H. lugubris* specimen was used for both the 3 × 3 × 2 µm and 10 × 10 × 2 µm substrates, and a different larva was used for the 3 × 3 × 4 µm surface.

## Estimating the elastic modulus of the microtrichia cuticle

Based on our observation that the microtrichia never showed any side contact, even when the suction discs were in contact with micro-structured substrates, we estimated the minimum elastic modulus of the microtrichia cuticle. We estimated the maximum force $F$ on one microtrichium, perpendicular to the surface, as 56 nN. This was obtained based on the following assumptions: (1) the suction discs are loaded with a peak normal force of 11.6 mN, ie, 1.9 mN per sucker (from centrifuge measurements of *H. lugubris*); (2) equal loading of ca 34,000 spine-like microtrichia in tip contact (based on the area of the suction disc bearing spine-like microtrichia of ca 49,000 µm², with the average microtrichia tip density of 0.7 per µm²). The microtrichia were assumed to be cylindrical, with a length $L = 6.7 \pm 0.5 \mu m$ and diameter $D = 0.56 \pm 0.01 \mu m$ (mean of means ± standard error of the mean; n = 2 *H. lugubris*), and the angle between the unloaded microtrichia and the surface was estimated as $\alpha = 45 \pm 3°$ (mean ±

SD measured from five microtrichia of *H. lugubris*). The local adhesion and friction force of the microtrichium were assumed to be negligible.

Following *Goss and Chaouki, 2016*, the elastic modulus below which a cylindrical beam loaded at an angle α (see $\phi_a$ definition) would exhibit side contact is

$$E = \left[ \frac{1}{K(p^2) - F(\phi_a | p^2)} \right]^2 \frac{FL^2}{I}$$

where $K\left(p^2\right)$ and $F\left(\phi_a | p^2\right)$ are the complete and incomplete elliptic integrals of the first kind, respectively, $p^2 = 1/2$ is the elliptic modulus, $\phi_a = sin^{-1}\left(\frac{sin\alpha/2}{p}\right)$ is the elliptic amplitude, and $I = D^4\pi/64$ is the second moment of area. See *Figure 6d–ii* for a schematic of a hypothetical scenario where microtrichia make side contacts on a smooth surface.

## Acknowledgements

We would like to thank P Ladurner for assisting in sample collection and providing laboratory space near to the field site and M Sutcliffe for helping with the surface profilometry measurements. We are also grateful to KH Muller and JN Skepper at the Cambridge Advanced Imaging Centre for their help in preparing and imaging SEM samples.

## Additional information

### Funding

| Funder | Grant reference number | Author |
| --- | --- | --- |
| EU Horizon 2020 research and innovation programme | No. 642861 | Victor Kang Walter Federle |

The funders had no role in study design, data collection and interpretation, or the decision to submit the work for publication.

### Author contributions

Victor Kang, Conceptualization, Data curation, Formal analysis, Investigation, Methodology, Project administration, Visualization, Writing – original draft, Writing – review and editing; Robin T White, Methodology, Writing – original draft, Writing – review and editing; Simon Chen, Methodology, Writing – review and editing; Walter Federle, Conceptualization, Funding acquisition, Methodology, Supervision, Writing – review and editing

### Author ORCIDs

Victor Kang  http://orcid.org/0000-0003-0959-1364
Robin T White  http://orcid.org/0000-0002-0030-2872
Simon Chen  http://orcid.org/0000-0002-6316-7567
Walter Federle  http://orcid.org/0000-0002-6375-3005

### Decision letter and Author response

Decision letter https://doi.org/10.7554/eLife.63250.sa1
Author response https://doi.org/10.7554/eLife.63250.sa2

## Additional files

### Supplementary files

• Transparent reporting form

### Data availability

Micro-computed tomography data is available in Dryad repository https://doi.org/10.5061/dryad.9zw3r22c2. Source data files have been provided for Figures 3, 4, and 5.

The following dataset was generated:

| Author(s) | Year | Dataset title | Dataset URL | Database and Identifier |
|---|---|---|---|---|
| Kang V | 2020 | Supplementary dataset for Extreme suction attachment performance from specialised insects living in mountain streams (Diptera: Blephariceridae) | https://doi.org/10.5061/dryad.9zw3r22c2 | Dryad Digital Repository, 10.5061/dryad.9zw3r22c2 |

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
