## [Decision Letter]

**Acceptance summary:**

Kang et al. eloquently describe the active suction organ that the larvae of aquatic insects of the dipteran family Blephariceridae use to adhere robustly to complex surfaces. While the morphology of the mechanism has been reported previously, its biomechanical adhesion function and performance across different substrates has been unknown. The authors present three advances. First, adhesion performance on rough, micro-rough, and smooth surfaces is quantified using an effective centrifugal setup. The performance tests show the larvae can resist shear forces up to 1100 times their body weight on smooth surfaces. Second, the suction function is visualized in vivo using interference reflection microscopy. This reveals that small hair-like microtrichia can enter gaps in the surface. Because the microtrichia are angled inward, the authors hypothesize that the microtrichia's angle and small size helps with interlocking and increasing suction-based attachment on rough surfaces. Finally, the adhesion performance of the Blephariceridae larvae is compared to other species, revealing that the weight-specific shear attachment on smooth surfaces is 3-10 times greater than found in stick insects. The finding that the larvae have such high attachment forces is impressive and the study offers new biological insights that may inspire engineers to invent new underwater suction mechanisms.

**Decision letter after peer review:**

Thank you for submitting your article "Extreme suction attachment performance from specialised insects living in mountain streams (Diptera: Blephariceridae)" for consideration by *eLife*. Your article has been reviewed by 2 peer reviewers, and the evaluation has been overseen by a Reviewing Editor and George Perry as the Senior Editor. The reviewers have opted to remain anonymous.

The reviewers have discussed the reviews with one another and the Reviewing Editor has drafted this decision to help you prepare a revised submission. Please comply with the comments and suggestions to your best ability, and respond in a point-by-point fashion. This is essential to enable the reviewing editor to fully evaluate the merit of your revision.

As the editors have judged that your manuscript is of interest, but as described below that additional work is required before it is published, we would like to draw your attention to changes in our revision policy that we have made in response to COVID-19 (https://elifesciences.org/articles/57162). First, because many researchers have temporarily lost access to the labs, we will give authors as much time as they need to submit revised manuscripts. We are also offering, if you choose, to post the manuscript to bioRxiv (if it is not already there) along with this decision letter and a formal designation that the manuscript is "in revision at *eLife*". Please let us know if you would like to pursue this option. (If your work is more suitable for medRxiv, you will need to post the preprint yourself, as the mechanisms for us to do so are still in development.)

Summary:

Kang et al. eloquently describe the active suction organ that the larvae of aquatic insects of the Dipterian family Blephariceridae use to adhere robustly to complex surfaces. While the morphology of the mechanism has been reported previously, it's biomechanical adhesion function and performance across different substrates is unknown. The authors present three advances. First, they quantify the adhesion performance on rough, micro-rough, and smooth surfaces using an effective centrifugal setup. The ultimate adhesion tests show the larvae can resist shear forces up to 1100 times their body weight on smooth surfaces. Second, they visualize the suction function in vivo using interference reflection microscopy. This reveals that small hair like microtrichia can enter gaps in the surface. Because the microtrichia are angled inward, the authors surmise that the microtrichia's angle and small size helps increase adhesion contact area on rough surfaces. Finally, they compare the adhesion performance of the Blephariceridae larvae to other species, showing it is 3-10 times greater than found in stick insects. The finding that the larvae have such high attachment forces is impressive and the study offers new biological insights that may inspire engineers to invent new underwater suction mechanisms.

Essential revisions:

Although the reviewers were generally appreciative of the well-written manuscript and the remarkable performance reported for the active suction mechanism, the consensus is that the mechanism itself is not described in sufficient detail for the reader to fully appreciate the advance. Hence the main critiques focus on helping the authors to further flesh out the mechanism and report it in more mechanistic detail like how other adhesion mechanism are described functionally across the biomechanical literature. Further the presentation of the figures does not meet graphic design clarity standards essential to inform *eLife*'s broad readership. To provide guidance, we list the following essential revisions.

1. The introduction states that the suction organs have been observed, however, the manuscript does not communicate the observed mechanism as one would expect in the biomechanical adhesion literature. Instead it reports the measurements of the force and a suggestion that the microtrichia may be involved. We were hoping to find a quantitative report of the mechanism integrating the force data and microscopy images into biomechanical diagrams and to the extent possible, equations, that capture and communicate the mechanism as quantitatively as possible. Whereas we are not requesting further measurements, because the performance of the mechanism is well documented, we do ask a more in-depth biomechanical analysis that spells out the mechanism in a way it can be compared to the other classic mechanisms that the authors compare to. If this requires some additional measurements to inform the model, those efforts would be well worth it. In case the authors can use a mechanistic analysis lead, we recommend reviewing a couple of papers. E.g. Jeffries, Lindsie, and David Lentink. "Design Principles and Function of Mechanical Fasteners in Nature and Technology." Applied Mechanics Reviews 72.5 (2020). Or any other review or research paper that the authors find more useful.

2. Please clarify if the experiments are done in air or underwater. We consider underwater as most appropriate; at minimum the surface should be wetted. The authors mention that the Stefan adhesion forces underwater would be higher than in air, but it's not clear if that statement pertains to the experiment. Please provide a full clarification, and in case the experiments were performed in air we would prefer to see them performed in water. If this is not possible, the manuscript should be entirely transparent on this matter so the reader can evaluate the precise merit of this study and its limitations fully.

3. We found the images confusing at times. To resolve this we would like to see clear schematics (avatars) that ground the reader's perspective in all figures.

4. Considering *eLife*'s broad multidisciplinary readership and the appeal of this study for bioinspired designers and engineers, Figure 1d,e has to provide better anatomical readability. Please assume a Biology and Engineering undergrad level for the first figure, ensuring all definitions and anatomical names can be fully comprehended without reference to other literature. Please provide clear connections to the different views and perspectives presented in the panels leveraging graphic design to the benefit of the interested reader not familiar with insect morphology.

5. Likewise, Figure 2 is also confusing. A schematic is in order to show the reader what they are looking at, how the images relate, and why they matter (significance) for understanding the main findings reported in this manuscript.

6. Figure 3 clearly shows that course-rough surfaces provide far less adhesion force. We wonder, are there any images similar to Figure 6 showing that the microtrichia cannot enter the gaps? To comprehend what causes the differences, we would like to see a report of the length scale of the microtrichia compared to that of the gap's dimensions, both for the rough and micro rough surfaces. To clarify this in a universal fashion, please consider reporting gap size non-dimensionally based on the relevant microtrichia length scale. More discussion of the relevant length scales would help bring the force measurements and the observations of the microtrichia together.

7. Figure 6 is an important figure, so it would help the reader to more easily grasp the viewing perspective using diagrams and avatars. I panel a, a schematic should clearly define the suction disc fringe and the perspective shown. What part is the suction disc and what is the length scale of this image compared to the suction disc? Also, it would be useful if the columns of the microstructure could all be aligned for clarity.

8. Currently, the authors provide an estimate of the shear stress. It would be helpful to also include the normal stress based on the normal force data on smooth surfaces for lugubris. It would be informative for the reader to know if it exceeds 1 atm. If so, that is a very interesting finding. Please report and discuss what you find in the revised manuscript.

9. Discussion: Please include a comparison of the magnitude of shear and normal stress that this suction mechanism creates with that of other organisms. Currently the comparison is done with force per body weight, which is biologically relevant. However, reporting stress provides an objective bio-mechanistic perspective on adhesion performance.

10. Discussion, Ln 300: The suggestion that the inward-facing microtrichia may function to prevent inward slipping of the suction cup is interesting. Please discuss the tradeoff between smooth and micro-rough surfaces: is it possible that on micro-rough surfaces the microtrichia are better able to resist slip, but on smooth surfaces, the seal is better? And if so, this would suggest the effect of a better seal is more important than preventing slip, since performance is better on smooth surfaces? In-vivo visualization during failure would be very informative (in future work).

11. Please discuss why there may be an intricate branching of the fan-fibres into the microtrichia. E.g. in the gecko, the branched tendons insert into the lamella, supporting the large tensile loads applied to the adhesive. However, here it is less clear if large tensile loads would be applied to the microtrichia. It seems logical that applying large normal loads to the suction cup should be done at its centre, resulting in decreased pressure if no slip occurs (as opposed to applying the normal force to the rim, which would not decrease pressure). So, this would not explain the intricate network of fan-fibres. However, for shear loads, it could make more sense: pulling in shear would engage the microtrichia on the far side of the cup, and the fan-fibres could help transmit this tension. It might be worth thinking this through and discussing the outcome in the paper to strengthen the mechanistic analysis.

12. We would be excited to learn if the authors have thoughts on the slight curvature of the microtrichia and how it may be involved in the adhesion mechanism. In case this is purely speculative, this could go into the last paragraph of the paper, alternatively it could go into the biomechanical model of the mechanism.

[Editors' note: further revisions were suggested prior to acceptance, as described below.]

Thank you for submitting your article "Extreme suction attachment performance from specialised insects living in mountain streams (Diptera: Blephariceridae)" for consideration by *eLife*. Your article has been reviewed by 2 peer reviewers, and the evaluation has been overseen by a Reviewing Editor and George Perry as the Senior Editor. The reviewers have opted to remain anonymous.

The reviewers have discussed their reviews with one another, and the Reviewing Editor has drafted this to help you prepare a revised submission. Please comply with the comments and suggestions to your best ability, and respond in a point-by-point fashion. This is essential to enable the reviewing editor to fully evaluate the merit of your revision. Upon evaluating your current revision, we noted these expectations were not fully met. To evaluate the revision, we had to request another review cycle, and this uncovered some remaining issues. We offer the authors this final chance to comply and transparently communicate the manuscript changes.

Essential revisions:

1) The videos and images are striking but difficult to interpret. A schematic or two of the entire suction disc organ would be particularly useful; with the various parts (piston, v-shaped notch) all labelled clearly. That will make it easier to find them in the SEM images, since the SEMS have several different views and many detail that obscure the main parts.

2) Please draw the microtrichia and supporting beams in the schematic to make the images in Figure 2 easier to interpret for *eLife*'s broad readership.

3) The authors compare normal to shear stress, but the reader wasn't informed about normal stress yet. The authors should clarify earlier that shear and normal directions will be tested. The justification why the study focusses on shear stress should be presented earlier to help the reader get the main point. For example, this could be accomplished by including a view of the larvae on a rock in a stream showing the natural conditions under which they might be dislodged. Even a schematic would do. Presenting this early, e.g. "Figure 1a", would make all the difference for *eLife*'s broad readership.

4) The authors model predicts that the microtrichia cuticle has a Young's modulus is 0.3 GPa. Please explain what the composition of the cuticle is and how the Young's modulus compares to the stiffness of similarly composed biomaterials.

5) The description of various microtrichia scenarios is confusing for the general reader. A diagram could help (or otherwise consider deleting this section). We were hoping to find a quantitative report of the mechanism integrating the force data and microscopy images into biomechanical diagrams and to the extent possible, equations, that capture and communicate the mechanism as quantitatively as possible. Ideally, the model would capture the basic results seen in the force data, for instance, explaining the change in force as surface roughness varied. In case this is outside the scope of the study, please discuss this open avenue of research in one of the closing sections of the manuscript. So, the reader is alerted that the mechanism isn't fully understood yet and which further research steps are needed to close this gap.

6) The authors should either show a clarifying schematic (preferred) or present a clear description of "beam side contact".

7) Figure 3 – the avatars are helpful; to fully clarify this figure, please draw a symbol to clearly indicate the axis of spin of the device as well.

8) The insets are very helpful. Many readers will, however, remain confused by the orientations of the images. Is the V-notch anterior or posterior? In 1b, it appears to be located on the left side of the discs, but in 1a, left appears to be anterior? Then in 2a, the inset shows the disc with what appears to be the V-notch located at the top of the inset, which is different than the V-C-D reference frame in the image? The fact that the slice is at 45 degrees, but then the reference frame seems to be parallel to the slice seems odd. What is most important for the reader is to know in 2a which direction is toward the center of the disc and which is toward the edge. It is not currently clear. Finally, in 2f, it now appears as though the disc has flipped orientations with respect to the inset? Is it possible to maintain the same orientation? Please resolve these issues holistically for the general *eLife* reader.

9) 6a-iii appears to be a different location than 6a-ii? If so, please add a schematic here as well to well-inform the general *eLife* reader. Please note 6-a-iii is missing a label.

10) Despite the additional explanation, the following two items that are unclear:

A -"To function effectively and to avoid buckling in this situation, interlocking structures…"

Buckling is usually defined as a sudden change in shape under an increasing load, often a beam loaded axially. Here there is a side load on the beam, so is this simply transverse bending rather than buckling? Or is the load on the side inducing buckling, please clarify what you would like to confer to the reader by thoughtfully considering the connotations.

B -"A second effect of the microtrichia curvature is that bending and thus side contact of the fine microtrichia tips is avoided, which would again reduce their ability to interlock with the substrate."

Is bending avoided via curvature? Or is it just that under a given amount of bending, having a pre-curve results in a shape that does not make side-contact? Please clarify so the general *eLife* reader can follow.

---

## [Author Response]

Essential revisions:Although the reviewers were generally appreciative of the well-written manuscript and the remarkable performance reported for the active suction mechanism, the consensus is that the mechanism itself is not described in sufficient detail for the reader to fully appreciate the advance. Hence the main critiques focus on helping the authors to further flesh out the mechanism and report it in more mechanistic detail like how other adhesion mechanism are described functionally across the biomechanical literature. Further the presentation of the figures does not meet graphic design clarity standards essential to inform eLife's broad readership. To provide guidance, we list the following essential revisions.1. The introduction states that the suction organs have been observed, however, the manuscript does not communicate the observed mechanism as one would expect in the biomechanical adhesion literature. Instead it reports the measurements of the force and a suggestion that the microtrichia may be involved. We were hoping to find a quantitative report of the mechanism integrating the force data and microscopy images into biomechanical diagrams and to the extent possible, equations, that capture and communicate the mechanism as quantitatively as possible. Whereas we are not requesting further measurements, because the performance of the mechanism is well documented, we do ask a more in-depth biomechanical analysis that spells out the mechanism in a way it can be compared to the other classic mechanisms that the authors compare to. If this requires some additional measurements to inform the model, those efforts would be well worth it. In case the authors can use a mechanistic analysis lead, we recommend reviewing a couple of papers. E.g. Jeffries, Lindsie, and David Lentink. "Design Principles and Function of Mechanical Fasteners in Nature and Technology." Applied Mechanics Reviews 72.5 (2020). Or any other review or research paper that the authors find more useful.

We thank the reviewers for highlighting this gap in our manuscript. After reading the suggested paper and reassessing our manuscript, we decided to include a new theoretical model and a schematic to further support our findings on the function of microtrichia during suction attachments (Figure 7, with accompanying text in Discussion lines 287-299, and Materials and methods lines 533 – 552). In order for the microtrichia to effectively interlock with surface asperities, they need to have sufficient stiffness to resist bending and buckling. Using our model, we estimated the elastic modulus of the microtrichia cuticle necessary to prevent side contact to be 0.3 – 0.4 GPa or higher, which is a range typical of stiff sclerotised cuticle, –which can have elastic moduli as high as 20 GPa. This result, along with our in vivo observations of the microtrichia always making tip contact on smooth and microstructured surfaces and microscopy images revealing their solid (filled-in) cuticular ultrastructure, strongly indicates that the spine-like microtrichia are stiff structures that can improve suction performance by interlocking their tips with surface asperities and increasing friction near the disc rim.

2. Please clarify if the experiments are done in air or underwater. We consider underwater as most appropriate; at minimum the surface should be wetted. The authors mention that the Stefan adhesion forces underwater would be higher than in air, but it's not clear if that statement pertains to the experiment. Please provide a full clarification, and in case the experiments were performed in air we would prefer to see them performed in water. If this is not possible, the manuscript should be entirely transparent on this matter so the reader can evaluate the precise merit of this study and its limitations fully.

Thank you for the comment. We have added this sentence to Line 116 to clarify the experimental condition:

“Each specimen was wetted with a droplet of water prior to centrifugation (see Materials and methods for details). Interference reflection microscopy (IRM, see below) observations showed that the contact of the suction organs under these conditions was completely wet, and no air bubbles were present in the contact zone.”

Please note that, since the centrifuge force measurement depends on the insect's (unsubmerged) weight and requires high quality video recording to determine the precise time of detachment, it is not possible to do these experiments entirely underwater. Instead, we wetted the blepharicerid larvae immediately prior to centrifugation, as described in the Materials and methods section (lines 465 468). The contact of the suction organs under these conditions was completely wet (IRM observations showed that no air bubbles were present in the contact zone) and the results are therefore comparable to the performance of the suction organs under natural conditions. Air bubbles in the contact zone would also be inconsistent with the lowest suction pressures we observed, as such air bubbles would expand so much in volume that the suction discs would likely detach. The wetting of the body surface did not influence the measurements via the body weight, as we could see small water droplets being removed from the larva during the centrifugation, and the insects were weighed after the experiment.

3. We found the images confusing at times. To resolve this we would like to see clear schematics (avatars) that ground the reader's perspective in all figures.

Thank you for raising this issue. We have revised Figures 1, 2, and 6 accordingly, and have included a new figure (Figure 7) that portrays a mechanical model of how the microtrichia system could interlock and avoid buckling against rough substrates (outlined in detail in the response to #1).

4. Considering eLife's broad multidisciplinary readership and the appeal of this study for bioinspired designers and engineers, Figure 1d,e has to provide better anatomical readability. Please assume a Biology and Engineering undergrad level for the first figure, ensuring all definitions and anatomical names can be fully comprehended without reference to other literature. Please provide clear connections to the different views and perspectives presented in the panels leveraging graphic design to the benefit of the interested reader not familiar with insect morphology.

We acknowledge that the previous layout of Figure 1 may have been confusing to non-specialists. We have revised Figure 1 by incorporating additional schematics and labels. In addition, please note that we included a Rich Media File (Video 1) during the initial submission that showcases the 3D rendering based on micro-CT data. Since Figure 1d and 1e are based on this dataset, we hope that this Rich Media File will further clarify the morphology to the reader.

5. Likewise, Figure 2 is also confusing. A schematic is in order to show the reader what they are looking at, how the images relate, and why they matter (significance) for understanding the main findings reported in this manuscript.

We have added a schematic of the suction disc to better orient the reader (see Figure 2). This is in addition to the changes made in Figure 1, and we believe this combination improves clarity for the reader.

6. Figure 3 clearly shows that course-rough surfaces provide far less adhesion force. We wonder, are there any images similar to Figure 6 showing that the microtrichia cannot enter the gaps? To comprehend what causes the differences, we would like to see a report of the length scale of the microtrichia compared to that of the gap's dimensions, both for the rough and micro rough surfaces. To clarify this in a universal fashion, please consider reporting gap size non-dimensionally based on the relevant microtrichia length scale. More discussion of the relevant length scales would help bring the force measurements and the observations of the microtrichia together.

Unfortunately, it is not possible to visualise the contact of the suction organs on the micro and coarse rough substrates since they are not transparent (the substrates have a polyester backing layer). To get around this problem, we used the epoxy-based microstructured substrates. The length-scales of the microtrichia (mean length 6.7 µm) and of the rough test surfaces (peak-to-valley heights: micro-rough 4.56 µm, coarse-rough 78.82 µm) are given in Table 1, Figure 7, and in the text. The ability of the suction organ to reach into the gaps between the asperities depends not only on the length of the microtrichia but also on the flexibility of the suction disc as a whole, as it can deform around the larger structures. For clarification, we have added an explanation to the Discussion (lines 226 – 229): "It is likely that both blepharicerid suction organs and stick insect adhesive pads are unable to cope with coarse surface roughness. The adhesive pads of both insects may be unable to fully mould to the large asperities, and the length of the microtrichia may be insufficient to reach the lower regions of the surface profile (Figure 6).”

In addition, we have revised Figure 6 to include schematics that clearly illustrate the length-scales of the microtrichia in comparison to the microstructured surfaces.

7. Figure 6 is an important figure, so it would help the reader to more easily grasp the viewing perspective using diagrams and avatars. I panel a, a schematic should clearly define the suction disc fringe and the perspective shown. What part is the suction disc and what is the length scale of this image compared to the suction disc? Also, it would be useful if the columns of the microstructure could all be aligned for clarity.

Thank you for the suggestions. We have revised the figure so that the sub-panels are aligned, and we have also added a schematic to highlight the approximate location and size of the images with respect to the overall suction disc (Figure 6a-ii). Moreover, we added scaled schematics to convey the length-scales of the microtrichia with respect to the microstructured surfaces.

8. Currently, the authors provide an estimate of the shear stress. It would be helpful to also include the normal stress based on the normal force data on smooth surfaces for lugubris. It would be informative for the reader to know if it exceeds 1 atm. If so, that is a very interesting finding. Please report and discuss what you find in the revised manuscript.

We have revised Table 2 (line 561) to include both shear and normal stress estimates for *L. cordata* and *H. lugubris*. The two highest normal stress values were obtained from two different *L. cordata* larvae, at 73 and 77 kPa. These are conservative estimates, however, as (1) we were unable to detach them using the centrifuge method, so the maximum recorded forces prior to termination were used; (2) we assumed the contact area to include the outer fringe layer; (3) all six organs were assumed to remain in contact prior to detachment. If we use a contact area without the fringe layer, then the normal stresses would be 108 and 114 kPa. In addition, larvae often attach with fewer than six organs immediately prior to detachment (also described Frutiger A. 2002. The function of the suckers of larval net-winged midges (Diptera: Blephariceridae). Freshwater Biology 47:293–302), which quickly increases the stress values (e.g., if three organs were in contact, then stress values would be 216 and 228 kPa). Although a pressure difference of 101 kPa (1 atm at standard sea level and temperature) is considered the upper threshold for suction attachments in air, this is not the case underwater, where the high cohesive strength of water allows for pressure differences to exceed 1 atm. For example, octopus and squid suckers can withstand normal stresses greater than 101 kPa, reaching up to 271 kPa for the octopus and 830 kPa for squids (Smith AM. 1996. Cephalopod sucker design and the physical limits to negative pressure. J Exp Biol 199:949–58). Thus, blepharicerid suction organs, like those of the cephalopods, are able to withstand strong normal stresses that exceed 101 kPa.

We have incorporated elements of this text in the Discussion (lines 250 – 260) to address points #8 and 9.

9. Discussion: Please include a comparison of the magnitude of shear and normal stress that this suction mechanism creates with that of other organisms. Currently the comparison is done with force per body weight, which is biologically relevant. However, reporting stress provides an objective bio-mechanistic perspective on adhesion performance.

Please refer to our response to point #8 and the revised Discussion (lines 250 – 260).

10. Discussion, Ln 300: The suggestion that the inward-facing microtrichia may function to prevent inward slipping of the suction cup is interesting. Please discuss the tradeoff between smooth and micro-rough surfaces: is it possible that on micro-rough surfaces the microtrichia are better able to resist slip, but on smooth surfaces, the seal is better? And if so, this would suggest the effect of a better seal is more important than preventing slip, since performance is better on smooth surfaces? In-vivo visualization during failure would be very informative (in future work).

We believe both an effective seal and an increase in friction help with suction performance. For example, studies that tested bio-inspired suction cup designs on rough substrates (Wang Y et al. 2017. A biorobotic adhesive disc for underwater hitchhiking inspired by the remora suckerfish. Sci Robot 2:eaan8072; Ditsche P, Summers A. 2019. Learning from Northern clingfish (Gobiesox maeandricus): Bioinspired suction cups attach to rough surfaces. Philos Trans R Soc B Biol Sci 374:1784) found that the combination of a soft sealing margin and friction-enhancing microstructures leads to peak attachment performance on rough substrates. We agree, however, that without an efficient seal, the loss of suction performance will likely outweigh the gain from having friction-enhancing structures. We have incorporated elements of this text into the Discussion (lines 310 – 314).

11. Please discuss why there may be an intricate branching of the fan-fibres into the microtrichia. E.g. in the gecko, the branched tendons insert into the lamella, supporting the large tensile loads applied to the adhesive. However, here it is less clear if large tensile loads would be applied to the microtrichia. It seems logical that applying large normal loads to the suction cup should be done at its centre, resulting in decreased pressure if no slip occurs (as opposed to applying the normal force to the rim, which would not decrease pressure). So, this would not explain the intricate network of fan-fibres. However, for shear loads, it could make more sense: pulling in shear would engage the microtrichia on the far side of the cup, and the fan-fibres could help transmit this tension. It might be worth thinking this through and discussing the outcome in the paper to strengthen the mechanistic analysis.

Thank you for suggesting the interesting idea that the fan-fibres could serve to transmit forces during shear loads on the suction disc, which would pull interlocked microtrichia radially outward (away from the centre of the suction disc). As the direction of the fan-fibres in the default (unstrained) condition is perpendicular to the surface, however, these fibres could only transmit forces parallel to the surface after a substantial outward displacement of the interlocked microtrichia. Given that the radial extension of the fan-fibre zone is relatively small (*ca* 20 µm, see Figure 2a), and that the fan-fibres are up to 20 µm long, we think that shear force transmission via the fan-fibres is unlikely.

One possible function of the branched fan-fibres is to coordinate microtrichia movement. Tension on the fan-fibres could be achieved via a raised hydraulic pressure within the fan-fibre space. A possible hint that microtrichia movement is activated via the fan-fibres is the observation that the microtrichia often flickered in our IRM video recordings. It is also possible, however, that this movement is caused by the flow of water in and out of the suction organ.

Another possibility is that the presence of the fan-fibres is a consequence of the development of the microtrichia, which involves the formation of long cytoplasmic extensions of epidermal cells (Rietschel P. 1961. Bau, Funktion und Entwicklung der Haftorgane der Blepharoceridae. Z Morph Ökol Tiere 50:239–265). Since little is known about the detailed development of the microtrichia or the suction organ in general, additional work is needed to explore this hypothesis.

As the above arguments are highly speculatory, we would like to refrain from including them in the manuscript.

12. We would be excited to learn if the authors have thoughts on the slight curvature of the microtrichia and how it may be involved in the adhesion mechanism. In case this is purely speculative, this could go into the last paragraph of the paper, alternatively it could go into the biomechanical model of the mechanism.

Thank you for your suggestion. We have added the following paragraph to the Discussion (lines 301 – 308):

“The densely packed microtrichia are both slightly curved and tapered, and their base is much thicker than the tip. […] A second effect of the microtrichia curvature is that bending and thus side contact of the fine microtrichia tips is avoided, which would again reduce their ability to interlock with the substrate.”

[Editors' note: further revisions were suggested prior to acceptance, as described below.]

Essential revisions:1) The videos and images are striking but difficult to interpret. A schematic or two of the entire suction disc organ would be particularly useful; with the various parts (piston, v-shaped notch) all labelled clearly. That will make it easier to find them in the SEM images, since the SEMS have several different views and many detail that obscure the main parts.

Thank you for your feedback. We have made extensive revisions to both Figure 1 and 2 to make them clearer to the general audience. In particular, we have included two additional sub-panels to Figure 1 (as well as a new video, Video 1).

In Figure 2, we have included a scanning electron micrograph giving an overview of the suction disc that is accompanied by a schematic of the suction disc, both of which are clearly labelled. We have also reoriented sub-panels so that the suction discs all face the same direction (see also our response to Comment #8). We hope these revisions will improve the overall reading experience.

2) Please draw the microtrichia and supporting beams in the schematic to make the images in Figure 2 easier to interpret for eLife's broad readership.

This has been addressed in the revised Figure 2a-i and ii.

3) The authors compare normal to shear stress, but the reader wasn't informed about normal stress yet. The authors should clarify earlier that shear and normal directions will be tested. The justification why the study focusses on shear stress should be presented earlier to help the reader get the main point. For example, this could be accomplished by including a view of the larvae on a rock in a stream showing the natural conditions under which they might be dislodged. Even a schematic would do. Presenting this early, e.g. "Figure 1a", would make all the difference for eLife's broad readership.

We have included an image (Figure 1a) and a video (Video 1) of the larvae in their natural habitat that demonstrate the need for powerful attachments to resist high shear forces. We agree that these additions provide more context to the readers and thank the reviewer for the suggestions.

In the previous version we used multiple terms to refer to adhesion force. We have now replaced "adhesion" or "adhesive force" with "normal force" where applicable to improve consistency and readability. Please note that we refer to both shear and normal forces before line 254: “Peak shear and

normal (adhesive) forces per body weight were measured on horizontal and vertical substrates, respectively (Figure 3a and b)” (see line 120 in the re-submitted version).

4) The authors model predicts that the microtrichia cuticle has a Young's modulus is 0.3 GPa. Please explain what the composition of the cuticle is and how the Young's modulus compares to the stiffness of similarly composed biomaterials.

We have included additional information on insect cuticle and comparable biomaterials in the revised paragraph (line 288 – 301).

5) The description of various microtrichia scenarios is confusing for the general reader. A diagram could help (or otherwise consider deleting this section). We were hoping to find a quantitative report of the mechanism integrating the force data and microscopy images into biomechanical diagrams and to the extent possible, equations, that capture and communicate the mechanism as quantitatively as possible. Ideally, the model would capture the basic results seen in the force data, for instance, explaining the change in force as surface roughness varied. In case this is outside the scope of the study, please discuss this open avenue of research in one of the closing sections of the manuscript. So, the reader is alerted that the mechanism isn't fully understood yet and which further research steps are needed to close this gap.

We added this description of the possible functions of the microtrichia shape (curvature and taper) in response to a comment from the first round of reviews. We believe this section can be removed as it is based on speculations.

In terms of a biomechanical model that can explain the suction organ attachment performance on various substrates: due to the highly complex morphology of the blepharicerid suction organ and the relative novelty of the study system, it is currently not possible to formulate a quantitative model to capture their performance on rough substrates. We acknowledge this limitation and have added the following sentence (line 307): “The detailed biomechanics of how microtrichia-covered blepharicerid suction organs produce a tight seal on rough surfaces and interlock with small substrate asperities is beyond the scope of this study and remains to be explored in future work.”

6) The authors should either show a clarifying schematic (preferred) or present a clear description of "beam side contact".

Thank you for the suggestion. We have added Figure 6d-i and d-ii to illustrate side contact, and have added the following sentence to line 561: “See Figure 6d-ii for a schematic of a hypothetical scenario where microtrichia make side contact on a smooth surface.”

7) Figure 3 – the avatars are helpful; to fully clarify this figure, please draw a symbol to clearly indicate the axis of spin of the device as well.

We have addressed this in the revised Figure 3.

8) The insets are very helpful. Many readers will, however, remain confused by the orientations of the images. Is the V-notch anterior or posterior? In 1b, it appears to be located on the left side of the discs, but in 1a, left appears to be anterior? Then in 2a, the inset shows the disc with what appears to be the V-notch located at the top of the inset, which is different than the V-C-D reference frame in the image? The fact that the slice is at 45 degrees, but then the reference frame seems to be parallel to the slice seems odd. What is most important for the reader is to know in 2a which direction is toward the center of the disc and which is toward the edge. It is not currently clear. Finally, in 2f, it now appears as though the disc has flipped orientations with respect to the inset? Is it possible to maintain the same orientation? Please resolve these issues holistically for the general eLife reader.

We acknowledge that the orientations in the previous version of Figure 1 and 2 could have been more consistent. We have addressed this issue with additional direction markers for Figure 1c and d, as well as re-orienting Figure 2 sub-panels so that both the images and the schematics consistently face the same direction (left being anterior).

9) 6a-iii appears to be a different location than 6a-ii? If so, please add a schematic here as well to well-inform the general eLife reader. Please note 6-a-iii is missing a label.

We have added a schematic to 6a-iii to illustrate the imaged region of the suction disc. We have also added a label.

10) Despite the additional explanation, the following two items that are unclear:A -"To function effectively and to avoid buckling in this situation, interlocking structures…"Buckling is usually defined as a sudden change in shape under an increasing load, often a beam loaded axially. Here there is a side load on the beam, so is this simply transverse bending rather than buckling? Or is the load on the side inducing buckling, please clarify what you would like to confer to the reader by thoughtfully considering the connotations.

We realise that this sentence (“To function effectively and to avoid buckling in this situation, interlocking structures…”) is confusing as we are not referring to the buckling of the microtrichia but to the buckling of the disc rim, which is a common mechanism of failure in underwater suction attachments (see line 264 in re-submitted version). We have revised the sentence to remove this ambiguity: “To interlock effectively, structures like the remora spinules need to be stiff and strong [38,39].”

B -"A second effect of the microtrichia curvature is that bending and thus side contact of the fine microtrichia tips is avoided, which would again reduce their ability to interlock with the substrate."Is bending avoided via curvature? Or is it just that under a given amount of bending, having a pre-curve results in a shape that does not make side-contact? Please clarify so the general eLife reader can follow.

We have opted to remove this section (line 309-317 in the reviewed version) as it was causing some confusion for the reviewers (see Comment #5). Furthermore, since we currently lack experimental support for our ideas, this section is mainly speculation based on morphology, and removing it will not impact the findings nor the overall story.